# Protein Multimer Structure Prediction via Prompt Learning

**Ziqi Gao**[1,2][*] **Xiangguo Sun**[3] **, Zijing Liu**[4]**, Yu Li**[4]**, Hong Cheng**[3]**, Jia Li**[1,2][†]

[1]Hong Kong University of Science and Technology (Guangzhou)
[2]Hong Kong University of Science and Technology
[3]The Chinese University of Hong Kong
[4]IDEA Research, International Digital Economy Academy

## Abstract

Understanding the 3D structures of protein multimers is crucial, as they play a vital role in regulating various cellular processes. It has been empirically confirmed that the multimer structure prediction (MSP) can be well handled in a step-wise assembly fashion using provided dimer structures and predicted protein-protein interactions (PPIs). However, due to the biological gap in the formation of dimers and larger multimers, directly applying PPI prediction techniques can often cause a *poor generalization* to the MSP task. To address this challenge, we aim to extend the PPI knowledge to multimers of different scales (i.e., chain numbers). Specifically, we propose **PromptMSP**, a pre-training and **Prompt** tuning framework for **M**ultimer **S**tructure **P**rediction. First, we tailor the source and target tasks for effective PPI knowledge learning and efficient inference, respectively. We design PPI-inspired prompt learning to narrow the gaps of two task formats and generalize the PPI knowledge to multimers of different scales. We provide a meta-learning strategy to learn a reliable initialization of the prompt model, enabling our prompting framework to effectively adapt to limited data for large-scale multimers. Empirically, we achieve both significant accuracy (RMSD and TM-Score) and efficiency improvements compared to advanced MSP models. The code, data and checkpoints are released at https://github.com/zqgao22/PromptMSP.

## 1 Introduction

Recent advances in deep learning have driven the development of AlphaFold 2 (AF2) (Jumper et al., 2021), a groundbreaking method for predicting protein 3D structures. With minor modifications, AF2 can be extended to AlphaFold-Multimer (AFM) (Evans et al., 2021) to predict the 3D structure of multimers (i.e., proteins that consist of multiple chains), which is fundamental in understanding molecular functions and cellular signaling of many biological processes. AFM has been verified to accurately predict the structures of multimers with small scales (i.e., chain numbers). However, its performance rapidly declines as the scale increases.

For multimer structure prediction (**MSP**), another research line (Esquivel-Rodríguez et al., 2012; Aderinwale et al., 2022; Inbar et al., 2005; Bryant et al., 2022) follows the idea of step-wise assembly (Figure 1A), where the assembly action indicates the protein-protein interaction (PPI). It sequentially expands the assembly size by adding a chain with the highest docking probability. The advantage of this step-wise assembly is that it can effectively handle multimers with large scales by enjoying the breakthrough in dimer structure prediction methods (Ganea et al., 2021; Wang et al., 2023; Ketata et al., 2023; Ghani et al., 2021; Luo et al., 2023; Chu et al., 2023; Evans et al., 2021).

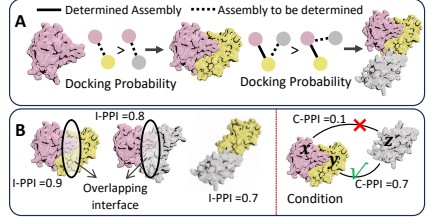

Figure 1: (**A**). Step-wise assembly for MSP. (**B**). Motivation for extending I-PPI to C-PPI.

---

[*]Work done during an internship at IDEA Research.
[†]Correspondence to: Jia Li (jialee@ust.hk).

As the most advanced assembly-based method, MoLPC (Bryant et al., 2022) applies independent PPI (I-PPI, i.e., both proteins are independent without the consideration of other proteins) to score the quality of a given assembly. Despite great potential, it does not consider important conditions in the assembly such as the influence of third-party proteins to PPI pairs. For example, in Figure 1B, if chain $x$ has already docked to chain $y$, the interface on $x$ that will contact with $z$ is partially occupied. Under this condition, the docking probability of $(x, z)$ may decrease to lower than that of $(y, z)$. We name this observation as condition PPI, or C-PPI. In short, neglecting the influence of C-PPI may easily lead to *poor generalization*. In this work, we focus on assembly-based MSP by learning C-PPI knowledge than I-PPI.

Learning effective C-PPI knowledge for MSP presents two main challenges. Firstly, we observe significant gaps in the C-PPI knowledge contained in multimers with varied scales (chain numbers), which suggests that the biological formation process of multimers may vary depending on their scales. Secondly, as shown in Figure 2, experimental structure data for large-scale multimers is extremely limited, making it even more difficult for the model to generalize them.

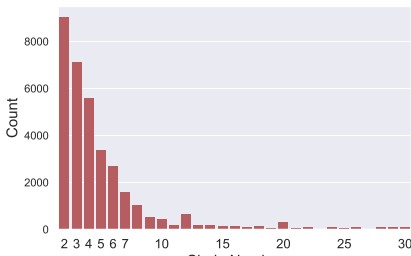

Figure 2: Distribution in chain numbers of multimers from the PDB database.

Recently, the rapidly evolving prompt learning (Liu et al., 2023; Sun et al., 2023a) techniques have shown promise to enhance the generalization of models to novel tasks and datasets. Inspired by this, a natural question arises: *can we prompt the model to predict C-PPIs for multimers with arbitrary scales*?

To address this, our core idea is to design learnable prompts to transform arbitrary-scale multimers into fixed-scale ones. Concretely, we first define the target task for training (tuning) the prompt model, which is conditional link prediction. Then, we additionally design the pre-training (source) task that learns to identify arbitrary assembled multimer's correctness. In the target task, we transform the two query chains to a virtual assembled multimer, which is input into the pre-trained model for the correctness score. We treat such a score as the linking probability of the query chains. **Therefore, arbitrary-scale prediction in the target task is reformulated as the fixed-scale one in the source task.**

Empirically, we investigate three settings: (1) assembly with ground-truth dimer structures to evaluate the accuracy of predicted docking path; (2) assembly with pre-computed dimers from AFM (Evans et al., 2021); and (3) assembly with pre-computed dimers from ESMFold (Lin et al., 2023). We show improved accuracy (in RMSD and TM-Score) and leading computational efficiency over recent state-of-the-art MSP baselines methods under these 3 settings. Overall, experiments demonstrate that our method has exceptional capacity and broad applicability.

## 2 RELATED WORK

**Multimer Structure Prediction.** Proteins typically work in cells in the form of multimers. However, determining the structures of multimers with biophysical experiments such as X-ray crystallography (Maveyraud & Mourey, 2020; Ilari & Savino, 2008) and cryogenic electron microscopy (Costa et al., 2017; Ho et al., 2020) can be extremely difficult and expensive. Recently, the deep learning (DL)-based AlphaFold-2 (Jumper et al., 2021) model can milestone-accurately predict protein structures from residue sequences. Moreover, recent studies have explored its potential for predicting multimer structures. However, they mostly require time-consuming multiple sequence alignment (MSA) operations and the performance significantly decreases for multimers with great chain numbers. Another research line assumes that the multimer structures can be predicted by adding its chains one by one. Multi-LZerD (Esquivel-Rodríguez et al., 2012) and RL-MLZerD (Aderinwale et al., 2022) respectively apply the genetic optimization and reinforcement learning strategies to select proper dimer structures for assembly. However, even when targeting only small-scale (3-, 4- and 5-chain) multimers, they still have low efficiency and are difficult to scale up for large multimers. By assuming that the dimer structures are already provided, MoLPC (Bryant et al., 2022) further simplifies this research line with the goal to predict just the correct docking path. With the help of

additional plDDT and dimer structure information, MoLPC has shown for the first time to predict the structures of large multimers with up to 30 chains.

**Prompt Learning for Pre-trained Models.** In the field of natural language processing (NLP), the prevailing prompt learning approach (Brown et al., 2020; Min et al., 2021) has shown gratifying success in transferring prior knowledge across various tasks. Narrowing the gap between the source and target tasks is important for the generalization of pre-trained models on novel tasks or data, which has not been fundamentally addressed with the pre-training-fine-tuning paradigm (Zhou et al., 2022). To achieve this, researchers have turned their attention to prompts. Specifically, a language prompt refers to a piece of text attached to the original input that helps guide a pre-trained model to produce desired outputs (Gao et al., 2020). Prompts can be either discrete or continuous (Sun et al., 2023b; Li et al., 2023). The discrete prompt (Gao et al., 2020; Schick & Schütze, 2020; Shin et al., 2020) usually refers to task descriptions from a pre-defined vocabulary, which can limit the flexibility of the prompt design due to the limited vocabulary space. In contrast, learnable prompts (Li & Liang, 2021; Zhang et al., 2021; Sun et al., 2023a) can be generated in a continuous space. Inspired by the success of prompt learning, we associate protein-protein interaction (PPI) knowledge (Kovács et al., 2019; Gao et al., 2023a), which is commonly present in multimers across various scales, to the pre-training phase. By fine-tuning only the prompt model, we can effectively adapt the PPI knowledge to the target task.

## 3 PRELIMINARIES

### 3.1 PROBLEM SETUP

**Assembly Graph.** We are given a set of chains (monomers), which is used to form a protein multimer. We represent a multimer with an assembly graph $\mathcal{G} = (\mathcal{V}, \mathcal{E})$. In $\mathcal{G}$, for the $i$-th chain, we obtain its chain-level embedding $c_i$ by the embedding function proposed in Chen et al. (2019). Each node $v_i \in \mathcal{V}$ can thus represent one chain with node attribute $c_i$. The assembly graph is an undirected, connected and acyclic (UCA) graph, with each edge representing an assembly action.

**Assembly Process.** For clarity, we apply an example in Figure 3 to illustrate the assembly process, which is achieved with the prepared dimer structures and the predicted assembly graph. Let us consider a multimer with 3 chains, whose undocked 3D structures are denoted as $X_1, X_2, X_3$. We consider an assembly graph with the edge set $\{(v_2, v_3), (v_3, v_1)\}$, and the dimer structures $\{(X_1^{12}, X_2^{12}), (X_1^{13}, X_3^{13}), (X_2^{23}, X_3^{23})\}$. First, we select the dimer of chains 2 and 3 as the start point, i.e., $X_2' = X_2^{23}$ and $X_3' = X_3^{23}$. Next, to dock the chain 1 onto chain 3, we compute the $SE(3)$ coordinate transformation $\mathbb{T}$ that aligns $X_3^{13}$ onto $X_3'$. Lastly, we apply $\mathbb{T}$ to $X_1$, resulting in the update coordinate $X_1'$ of chain 1.

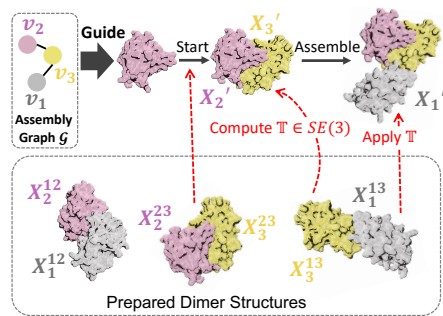

Figure 3: Assembly process with the predicted assembly graph and prepared dimers.

**Definition 1 (Assembly Correctness)** *For an $N$-chain multimer with a 3D structure $X$, its chains are represented by the nodes of an assembly graph $\mathcal{G}$. The assembly correctness $F(\mathcal{G}, X)$ is equivalent to the TM-Score (Zhang & Skolnick, 2004) between the assembled multimer and the ground-truth.*

With the above definitions, our paper aims to predict assembly graphs that maximize the TM-Scores, taking as inputs the residue sequences of chains and pre-calculated dimer structures.

### 3.2 SOURCE AND TARGET TASKS

In this paper, we adopt a pre-training (*source task*) and prompt fine-tuning (*target task*) framework to address the MSP problem. We consider two points for task designs: 1) With given multimers for pre-training, the model benefits from common intrinsic task subspace between the source and target task. 2) The target task should be designed to effectively learn the condition PPI (C-PPI) knowledge and efficiently complete the MSP inference.

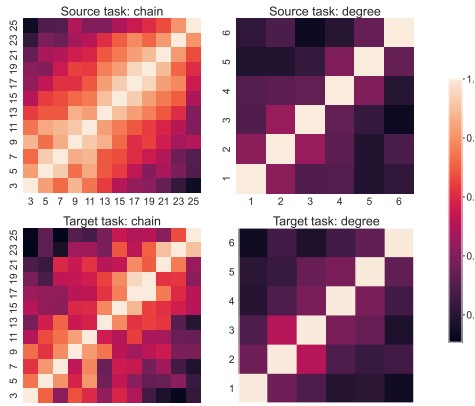

Figure 4: **Analysis on multimers with varied chain numbers.** We select some samples for evaluation and visualize heatmaps that show the similarity of the sample embeddings obtained from different pre-trained models. Each value on the axis suggests that the model is trained on data with the specific chain number or degree value. For example in the heatmap titled 'Source task: chain', the darkness at the [5,7] block represents the similarity between the embeddings extracted from two models that are trained under the source task with 5- and 7-chain multimers, respectively.

**Definition 2 (Source Data $\mathcal{D}_{sou}$)** *Each data instance in $\mathcal{D}_{sou}$ involves an assembly graph $\mathcal{G}_{sou}$ and a continuous label $y_{sou}$, i.e., $(\mathcal{G}_{sou}, y_{sou}) \in \mathcal{D}_{sou}$. For an $N$-chain multimer, $\mathcal{G}_{sou}$ is randomly generated as its $N$-node UCA assembly graph, and $y_{sou}$ is the assembly correctness.*

**Definition 3 (Target Data $\mathcal{D}_{tar}$)** *Each data instance in $\mathcal{D}_{tar}$ involves an assembly graph $\mathcal{G}_{tar} = (\mathcal{V}_{tar}, \mathcal{E}_{tar})$, an indicated node $v_d \in \mathcal{V}_{tar}$, an isolated node $v_u \notin \mathcal{G}_{tar}$ and the continuous label $y_{tar}$, i.e., $(\mathcal{G}_{tar}, v_d, v_u, y_{tar}) \in \mathcal{D}_{tar}$. For an $N$-chain multimer, $\mathcal{G}_{tar}$ is defined as its $(N-1)$-chain assembly graph. $y_{tar}$ is calculated by $y_{tar} = F((\{\mathcal{V}_{tar} \cup v_u\}, \{\mathcal{E}_{tar} \cup v_d v_u\}), \boldsymbol{X})$.*

**Source Task.** We design the source task as the *graph-level regression task*. Based on source data defined in Def. 2, the model is fed with a UCA assembly graph and is expected to output the continuous correctness score between 0 and 1. Note that theoretically, we can generate $N^{N-2}$ different UCA graphs and their corresponding labels with an $N$-chain multimer. This greatly broadens the available training data, enhancing the effectiveness of pre-training in learning MSP knowledge.

**Target Task.** We design the target task as the *link prediction task* (i.e., predicting the C-PPI probability). Based on target data defined in Def. 3, the target task aims to predict the presence of the link between nodes $v_d$ and $v_u$, which represent the docked and undocked chains, respectively.

We provide the detailed process to generate $\mathcal{D}_{sou}$ and $\mathcal{D}_{tar}$ in Appendix A.1. Overall, the source and target tasks learn assembly knowledge globally and locally, respectively. Unfortunately, multimers of varied scales exhibit distribution shifts in the target task, preventing the direct use of it for MSP. Next, we will empirically verify the existence of these shifts and their influence on MSP.

### 3.3 GAPS BETWEEN MULTIMERS WITH VARIED SCALES

We have presented a significant imbalance in the data of multimers (see Figure 2). Here, we analyze the gap in MSP knowledge among multimers with various scales (i.e., chain numbers), which can further consolidate our motivation of utilizing prompt learning for knowledge transfer. We also offer explanations for the reasons behind the gaps based on empirical observations.

We begin by analyzing the factor of chain number. We randomly select multimers for evaluation and divide the remaining ones into various training sets based on the chain number, which are then trained with independent models. We obtain the chain representations of the evaluating samples in each model. Lastly, we apply Centered Kernel Alignment (CKA) (Raghu et al., 2021), a function for evaluating representation similarity, to quantify the gaps in knowledge learned by any two models. We show the heatmaps of CKA in the Figure 4 and have two observations. (a) Low similarities are shown between data with small and large scales. (b) Generally, there is a positive correlation between the C-PPI knowledge gap and the difference in scales. In short, C-PPI knowledge greatly depends on the multimer scale.

To further explain these gaps, we re-divide the training sets based on the degree (i.e., neighbor number of a node) of assembly graphs and perform additional experiments. Specifically, we define the degree value as the highest node degree within each graph $\mathcal{G}_{sou}$ in the source task, and as the

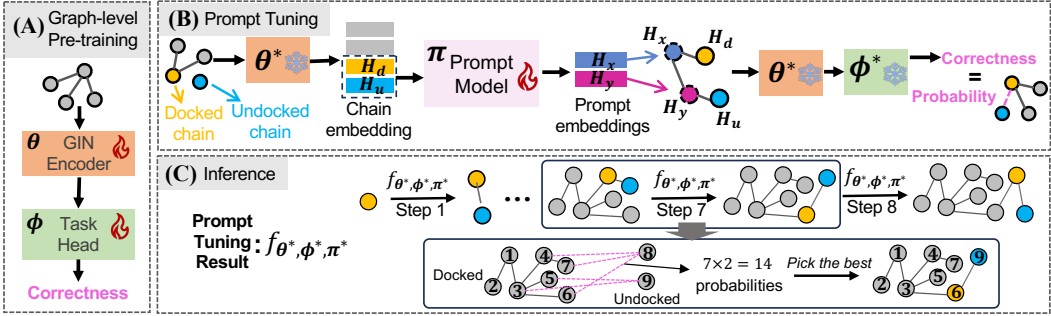

Figure 5: **The overview of PromptMSP. (A).** Firstly, we pre-train the GIN encoder and the task head under the graph-level regression task. After pre-training, given an arbitrary graph, $\theta^*$ and $\phi^*$ jointly output the correctness. **(B).** During prompt tuning, the prompt model takes embeddings of a pair of docked and undocked (query) chains as input and learns to produce prompt embeddings which form the entire 4-node path. $\theta^*$ and $\phi^*$ then jointly predict the correctness, which is equivalent to the linking probability. We use $f_{\theta^*,\phi^*,\pi^*}$ to denote the well trained pipeline that outputs the linking probability of query chains with target data instance as input. **(C).** If the target multimer has 9 chains, we sequentially perform 8 steps for inference. In each step, we use the well trained pipeline to calculate the probabilities for all possible chain pairs and select the most possible pair to assemble.

degree of node $v_d$ to be docked on in the target task. As shown in Figure 4, CKA heatmaps indicate that training samples with different degrees exhibit the gap in knowledge, which becomes even more significant than that between data with varying chain numbers. As observed, we conclude that the gap in chain numbers between data may be primarily due to the difference in degree of assembly graphs. Therefore, we intuitively associate the degree with the biological phenomenon of competitive docking that may occur in MSP, as evidenced by previous studies (Chang & Perez, 2022; Yan et al., 2020; Chang & Perez, 2023). In other words, multimers with more chains are more likely to yield assembly graphs with high degrees, and consequently, more instances of competitive docking. We expect that prompt learning can help bridge this knowledge gap.

# 4 PROPOSED APPROACH

**Overview of Our Approach.** Our approach is depicted in Figure 5, which follows a pre-training and prompt tuning paradigm. Firstly, using abundant data of small-scale (i.e., $3 \leq N \leq 5$) multimers, we pre-train the graph neural network (GNN) model on the source graph regression task. We then design the learnable prompt model, which can reformulate the conditional link prediction (target) task to the graph-level regression (source) task. In the process of task reformulation, an arbitrary-scale multimer in the target task is converted to a fixed-scale (i.e., $N = 4$) multimer in the source task. For inference, an $N$-chain multimer will go through $N - 1$ steps to be fully assembled. In each step, our model predicts the probabilities of all possible conditional links and selects the highest one to add a new chain. Besides, to further enhance the generalization ability, we provide a meta-learning strategy in Appendix C.5.

## 4.1 PRE-TRAINING ON THE SOURCE TASK

We apply graph neural network (GNN) architecture (Xu et al., 2018; Veličković et al., 2017; Tang et al., 2023) for graph-level regression (Cheng et al., 2023; Gao et al., 2023b). Our model first computes the node embeddings of an input UCA assembly graph using Graph Isomorphism Network (GIN, Xu et al. (2018)) to achieve its state-of-the-art performance. Kindly note that we can also apply other GNN variants (Kipf & Welling, 2016; Veličković et al., 2017; Tang et al., 2022; Liu et al., 2024; Li et al., 2019) for pre-training. Following Def. 2, we construct source data $\mathcal{D}_{sou}$ by using the oligomer (i.e., small-scale multimer) data only. The pre-training model approximates the assembly correctness with data instances of $\mathcal{D}_{sou}$:

$$\tilde{y}_{sou} = \text{GNN}(\mathcal{G}_{sou}; \theta, \phi) \approx F(\mathcal{G}_{sou}; \boldsymbol{X}) = y_{sou}, \quad (1)$$

where GNN represents combination of a GIN with parameters $\theta$ for obtaining node embeddings, a ReadOut function after the last GIN layer, and a task head with parameters $\phi$ to yield the prediction $\tilde{y}_{sou}$. As defined in Def. 1, $F$ represents the assembly correctness function for computing TM-Score between the assembled structure and the ground-truth (GT) structure $\boldsymbol{X}$.

We train the GNN by minimizing the discrepancy between predicted and GT correctness values:

$$\theta^*, \phi^* = \arg\min_{\theta,\phi} \sum\nolimits_{(\mathcal{G}_{sou}, y_{sou}) \in \mathcal{D}_{sou}} \mathcal{L}_{pre}(y_{sou} = F(\mathcal{G}_{sou}; \boldsymbol{X}), \tilde{y}_{sou} = \text{GNN}(\mathcal{G}_{sou}; \theta, \phi)), \quad (2)$$

where $\mathcal{L}_{pre}$ is the mean absolute error (MAE) loss function. After the pre-training phase, we obtain the pre-trained **GIN encoder** and the **task head** parameterized by $\theta^*$ and $\phi^*$, respectively.

## 4.2 ENSURING CONSISTENCY BETWEEN SOURCE AND TARGET TASKS

**Reformulating the Target Link Prediction Task.** The inference of an $N$-chain multimer under the source task setting requires all of its $N^{N-2}$ assembly graphs and their corresponding correctness from the pre-trained model. Therefore, when dealing with large-scale multimers, such inference manner requires effective UCA graph traversal algorithms and significant computational resources. The target task proposed in Section 3.2 can address this issue, which aims to predict the link presence (probability) between a pair of docked and undocked chains. As shown in Figure 5C, we can inference the structure of an $N$-chain multimer with just $N-1$ steps. At each step, we identify the most promising pair of (docked and undocked) chains.

The success of traditional pre-training and fine-tuning paradigm is due to that source and target tasks having a common task subspace, which allows for unobstructed knowledge transfer (Sun et al., 2023a). However, in this paper, source and target tasks are naturally different, namely graph-level and edge-level tasks, respectively. Here, we follow three principles to reformulate the target task: (1) To achieve consistency with the source task, the target task needs to be reformulated as a graph-level problem. (2) Due to the distribution shifts in multimers with varied chain numbers (Figure 4), a multimer of arbitrary scale in the target conditional link prediction task should be reformulated into a fixed-scale one in the source task. (3) The pre-trained GNN model is expected to effectively handle multimers of such "fixed-scale" in the source task. The upcoming introduction of prompt design will indicate that the fixed-scale value is 4. Therefore, to ensure (3), we limit the data used for pre-training to only multimer of $3 \leq N \leq 5$.

**Prompt Design.** Following Def. 3, we create the target data $\mathcal{D}_{tar}$ for prompt tuning. For clarity, we denote each data instance as a tuple form $(\mathcal{G}_{tar}, v_d, v_u, y_{tar}) \in \mathcal{D}_{tar}$, where $\mathcal{G}_{tar}$ denotes the current assembled multimer (i.e., condition), $v_d$ is a query chain within $\mathcal{G}_{tar}$ and $v_u$ is another query chain representing the undocked chain. We compute the last layer embeddings $H$ of all nodes in $\mathcal{G}_{tar}$ and the isolated $v_u$ with the pre-trained GIN encoder. To enable communications between target nodes $v_d$ and $v_u$, the prompt model parameterized by $\pi$ contains multiple cross atten-

$$H \in \mathbb{R}^{(|\mathcal{V}_{tar}|+1) \times d} = \theta^*(\mathcal{V}_{tar} \cup v_u, \mathcal{E}_{tar}), \quad (3)$$

$$H_x = \sigma_\pi \left( \left[ \text{softmax} \left( H_d^\top H_u \right) H_u^\top \right]^\top \right), \quad (4)$$

$$H_y = \sigma_\pi \left( \left[ \text{softmax} \left( H_u^\top H_d \right) H_d^\top \right]^\top \right), \quad (5)$$

$$\mathcal{G}_{pro} = (\mathcal{V}_{pro} = \{v_d, v_u, v_x, v_y\}, \\ \mathcal{E}_{pro} = \{e_{dx}, e_{xy}, e_{yu}\}), \quad (6)$$

$$\tilde{y}_{tar} = \phi^*(\theta^*(\mathcal{G}_{pro})), \quad (7)$$

tion layers (Vaswani et al., 2017; Wang et al., 2023) that map $H_u, H_d \in \mathbb{R}^d$ to vectors $H_x, H_y \in \mathbb{R}^d$, which represent the initial features of nodes $v_x$ and $v_y$. Finally, the pre-trained model outputs the assembly correctness of the 4-node prompt graph $\mathcal{G}_{pro}$. The whole target task pipeline of our method is represented by the equations on the right side.

Specifically, $\theta^*$ is the pre-trained GIN encoder, $\phi^*$ is the pre-trained task head and $d$ denotes the dimension of features. The prompt model $\pi$, which outputs a vector $\mathbb{R}^d$, includes non-trainable cross attention layers and the parametric function (Multi-Layer Perceptron, MLP) $\sigma_\pi$. Moreover, we use $f_{\theta^*, \phi^*, \pi^*}$ to **represent the entire pipeline** (Figure 5B) which takes input $(\mathcal{G}_{tar}, v_d, v_u)$ and outputs $\tilde{y}_{tar}$. A more detailed model architecture is shown in Appendix A.2.

**Prompt Design Intuition.** First of all, the link between two query chains is equivalent to the protein-protein interaction (PPI) in biology. We introduce the $\ell = 3$ path (Kovács et al., 2019; Yuen & Jansson, 2023), a widely validated biological rule for modeling PPI probabilities. Figure 6 describes the $\ell = 3$ rule, which is based on the fact that docking-based PPI generally requires proteins to have complementary surface representations for contact. It claims that the PPI probability of any two chains is not reflected by the number of their common neighbors (a.k.a., triadic closure principle (Lou et al., 2013; Sintos & Tsaparas, 2014)), but rather by the presence of a path of length $\ell = 3$. **In short, if there exists a 4-node path with query chains at the ends, they are highly likely to have a PPI (link).**

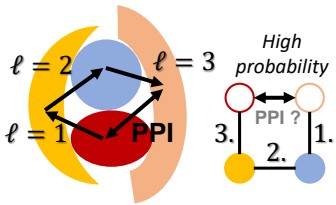

Figure 6: The $\ell = 3$ PPI rule.

Regardless of the node number of $\mathcal{G}_{tar}$, we treat the pre-trained model output based on this 4-node $\mathcal{G}_{pro}$ as the linking probability between $v_d$ and $v_u$. Unlike most of exsiting prompt learning techniques, the proposed task reformulation manner is naturally interpretable. Let us assume that these two query chains are highly likely to have a link, it will be reasonable to find two virtual chains that help form a valid $\ell = 3$ path. This also suggests that the assembly of $\mathcal{G}_{pro}$ tends to be correct, i.e., $\phi^*(\theta^*(\mathcal{G}_{pro})) \rightarrow 1$. Therefore, intuitively, the correctness of $\mathcal{G}_{pro}$ fed to the pre-trained model implies the linking probability between $v_u$ and $v_d$.

## 4.3 INFERENCE PROCESS WITH THE PROMPTING RESULT $f_{\pi^*|\theta^*,\phi^*}$

With prompt tuning, we obtain the whole framework pipeline $f_{\pi^*|\theta^*,\phi^*}$. For inference on a multimer with $N$ chains, we perform $N - 1$ assembly steps, in each of which we apply the *pipeline* $f_{\pi^*|\theta^*,\phi^*}$ to predict the linking probabilities of all pairs of chains and select the most likely pair for assembly.

## 5 EXPERIMENTS

**Datasets.** We collect all publicly available multimers ($3 \leq N \leq 30$) from the Protein Data Bank (PDB) database (Berman et al., 2000) on 2023-02-20. Referring to the data preprocessing method in MoLPC (Bryant et al., 2022), we obtain a total of 9,254 multimers. To further clarity, we use the abbreviation **PDB-M** to refer to the dataset applied in this paper. Overall, the preprocessing method ensures that PDB-M contains *high resolution, non-redundant multimers and is free from data leakage* (i.e., no sequences with a similarity greater than 40% between training and test sets). Due to the commonly low efficiency of all baselines, we define a data split for $3 \leq N \leq 30$ with a small test set size to enable comparison. Specifically, we select 10 for each scale of $3 \leq N \leq 10$, and 5 for each scale of $11 \leq N \leq 30$. Moreover, for comprehensive evaluation, we re-split the PDB-M dataset based on the release date of the PDB files to evaluate our method. Detailed information about the data preprocessing methods and the data statistic of split is in Appendix B.

**Baselines and Experimental Setup.** We compare our PROMPTMSP method with recent deep learning (DL) models and traditional software methods. For DL-based state-of-the-art methods, RL-MLZerd (Aderinwale et al., 2022) and AlphaFold-Multimer (AFM) (Evans et al., 2021) are included. For software methods, Multi-LZerd (Esquivel-Rodríguez et al., 2012) and MoLPC (Bryant et al., 2022) are included.

Since assembly-based methods require given dimers, we first use the ground-truth (GT) dimer structures (represented as **GT Dimer**) to evaluate the assembled multimer structures. For the pair of chains with contact, **GT Dimer** includes their native dimer structure drawn from the GT multimer. For those without contact, we use EquiDock (Ganea et al., 2021) for outputting dimer structures due to its fast inference speed. Moreover, since GT dimers are not always available, for pratical reasons, we consider to prepare dimers with AFM (Evans et al., 2021) (**AFM Dimer**) and ESM-Fold (Lin et al., 2023) (**ESMFold Dimer**). For baselines not requiring given dimers, we use these three kinds of dimers to reassemble based on the docking path mined in their predicted multimer, which is referred to as the † version of the baselines. Our experiments consist of 3 settings: 1) Since most baselines can not handle multimers with chain numbers $N > 10$. We follow **GT Dimer**, **AFM Dimer** and **ESMFold Dimer** to evaluate all baselines on the small-scale multimers ($3 \leq N \leq 10$) in the test set. 2) We evaluate MoLPC and our method by using these three types of dimers on the

Table 1: **Multimer structure prediction results.** Methods are evaluated on the test set of $3 \leq N \leq 10$ by using three types of pre-computed dimers. The test set includes 80 multimer samples in total (10 samples for each scale). For each dimer type and metric, the best method is **bold** and the second best is underlined. † represents the reassembly version of baselines. R(Avg): average RMSD; R(Med): median RMSD; T(Avg): average TM-Score; T(Med): median TM-Score.

| Methods | GT Dimer | | AFM Dimer | | ESMFold Dimer | |
|---|---|---|---|---|---|---|
| | R(Avg)/R(Med) | T(Avg)/T(Med) | R(Avg)/R(Med) | T(Avg)/T(Med) | R(Avg)/R(Med) | T(Avg)/T(Med) |
| Multi-LZerD | 31.50 / 33.94 | 0.28 / 0.25 | 31.50 / 33.94 | 0.28 / 0.25 | 31.50 / 33.94 | 0.28 / 0.25 |
| Multi-LZerD† | 18.90 / 19.30 | 0.54 / 0.38 | 29.68 / 27.96 | 0.30 / 0.33 | 33.00 / 31.07 | 0.25 / 0.29 |
| RL-MLZerD | 31.04 / 27.44 | 0.29 / 0.32 | 31.04 / 27.44 | 0.29 / 0.32 | 31.04 / 27.44 | 0.29 / 0.32 |
| RL-MLZerD† | 17.77 / 17.69 | 0.51 / 0.53 | 28.57 / 26.20 | 0.30 / 0.35 | 27.76 / 32.91 | 0.32 / 0.25 |
| AFM | 20.99 / 24.76 | 0.47 / 0.42 | 20.99 / 24.76 | 0.47 / 0.42 | **20.99 / 24.76** | **0.47 / 0.42** |
| AFM† | 16.79 / 16.02 | 0.59 / 0.59 | 18.98 / 19.05 | 0.50 / 0.48 | 26.76 / 29.95 | 0.33 / 0.30 |
| MoLPC | 18.53 / 18.08 | 0.52 / 0.55 | 23.06 / 23.92 | 0.43 / 0.42 | 30.17 / 29.45 | 0.31 / 0.31 |
| **PromptMSP** | **13.57 / 11.74** | **0.67 / 0.71** | **17.36 / 17.09** | **0.55 / 0.56** | 22.55 / 24.85 | 0.45 / 0.37 |

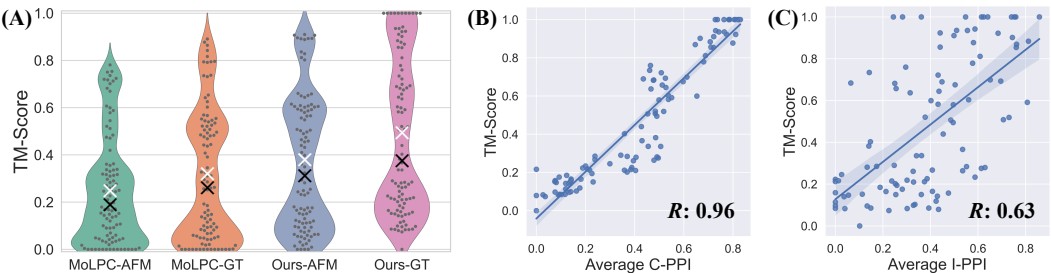

Figure 7: **(A).** TM-Score distribution of MoLPC and our method tested on multimers of $10 \leq N \leq 30$. The mean and median values are marked with white and black '×', respectively. **(B).** The relationship of learned C-PPI knowledge and the actual TM-Score. **(C).** The relationship of I-PPI learned by MoLPC and the actual TM-Score. We show the Pearson's correlation **$R$** for both.

entire test set ($3 \leq N \leq 30$). 3) We additionally split the PDB-M dataset based on the release date of multimers to evaluate the generalization ability of our method. We run all methods on 2 A100 SXM4 40GB GPUs and consider exceeding the memory limit or the resource of 10 GPU hours as failures, which are padded by the upper bound performance of all baselines.

**Evaluation Metrics.** To evaluate the performance of multimer structure prediction, we calculate the root-mean-square deviation (**RMSD**) and the **TM-score** both at the residue level. We report the mean and median values of both metrics.

**Multimer Structure Prediction Results.** Model performance on multimers of two kinds of scales ($3 \leq N \leq 10$, $11 \leq N \leq 30$) are summarized in Table 1 and Figure 7A, respectively. For small-scale multimers, our model achieves state-of-the-art on all metrics. In addition, we find that most MSP methods can benefit from the reassembly of GT or AFM dimer structures. Notably, our model can significantly outperform MoLPC, even though it does not require additional plDDT information or coarse information for protein interactions. For larger-scale multimers, our model also outperforms MoLPC, and outputs completely accurate prediction results for certain samples (i.e., TM-Score = 1.0 under `GT Dimer`). As for the failed inference samples of MoLPC, we relax the model's penalty term to successfully obtain the predictions instead of simply considering its TM-Score as 0. Despite this, our model can still achieve significant improvements under `GT Dimer`, `AFM Dimer` and `ESMFold Dimer`. The experimental results under the data split based on the release dates is in Appendix C.1.

Table 2 shows the inference efficiency of all baselines. As assembly-based methods require given dimer structures, we report the separate running time for predicting the docking path and preparing dimers, as well as the total time consumption. Kindly note that during inference, our method predicts the docking path without the need for pre-computed dimers. Therefore, to predict the structure of an $N$-chain multimer, our method (always) requires $N-1$ pre-computed dimers. We note that regardless of the dimer type used, our method is significantly faster than the other baselines. Our method also achieves higher efficiency in predicting the docking path compared to MoLPC. We provide more docking path inference results of our method (in Figure 9 in Appendix). We can find that as the scale increases, the inference time for a single assembly process (the orange curve) of our method does not increase, which suggests that the applicability of our model is not limited by the scale.

Table 2: Efficiency comparison (average MSP inference time).

| Time(min) | $3 \leq N \leq 10$ | | | $11 \leq N \leq 30$ | | |
|---|---|---|---|---|---|---|
| | Path | Dimer | Total | Path | Dimer | Total |
| Multi-LZerD | 187.51 | – | 187.50 | | | |
| RL-MLZerD | – | 173.88 | 173.88 | | – | |
| AFM | – | – | 155.72 | | | |
| MoLPC | 11.64 | 165.73 | 177.37 | 11.64 | 354.23 | 365.87 |
| **Ours**-GT | 0.01 | – | 0.01 | 0.04 | – | 0.04 |
| **Ours**-AFM | 0.01 | 80.79 | 80.80 | 0.04 | 187.44 | 187.48 |
| **Ours**-ESMFold | 0.01 | 0.35 | 0.36 | 0.01 | 1.09 | 1.10 |

**Ablation Study.** We perform ablation study in Table 3 to explore the significance of the prompt model and the C-PPI modelling strategy. If we remove the prompt model and apply the link prediction task both for pre-training and fine-tuning, the performance will greatly decrease by about 21.6% on large-scale multimers. This implies the contribution of prompt in unifying the C-PPI knowledge in multimers of different scales. Similarly, the significance of applying C-PPI modelling can be further illustrated through its relationship with the MSP problem. Figure 7(BC) indicates that I-PPI will bring negative transfer to the MSP task, ultimately hurting the performance.

Table 3: Ablation study with GT dimers.

| Prompt | C-PPI | $3 \leq N \leq 10$ | $11 \leq N \leq 30$ |
|---|---|---|---|
| × | ✓ | 0.55(-17.9%) | 0.29(-21.6%) |
| ✓ | × | 0.54(-19.4%) | 0.33(-10.8%) |
| ✓ | ✓ | 0.67 | 0.37 |

In Figure 8, we show the generalization ability of our method. The term 'w/o prompt' refers to the direct use of GNNs for conditional link prediction for MSP. We find that when introducing training multimers with the scale (e.g., $N > 11$) differs significantly from the testing multimers (i.e., $N = 7$), the performance of the 'w/o prompt' method notably declines. Conversely, for PROMPTMSP, adding arbitrary-scale multimers to the training set will improves the model's generalization ability. This indicates that our

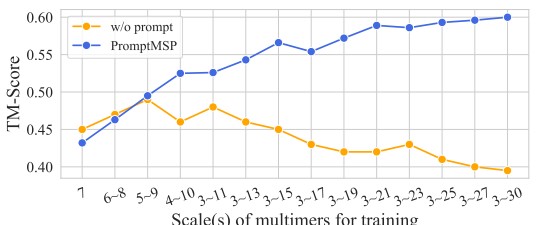

Figure 8: Results tested on $N$=7. We train our model and its 'w/o prompt' version on multimers of varied scale ranges.

model can effectively capture shared knowledge between varied-scale multimers, while blocking the knowledge gaps caused by distribution shifts.

# 6 CONCLUSION

Fast and effective methods for predicting multimer structures are essential tools to facilitate protein engineering and drug discovery. We follow the setting of sequentially assembling the target multimer according to the predicted assembly actions for multimer structure prediction (MSP). To achieve this, our main goal is to learn conditional PPI (C-PPI) knowledge that can adapt to multimers of varied scales (i.e., chain numbers). The proposed pre-training and prompt tuning framework can successfully narrow down the gaps between different scales of multimer data. To further enhance the adaptation of our method when facing data insufficiency, we introduce a meta-learning framework to learn a reliable prompt model initialization, which can be rapidly fine-tuned on scarce multimer data. Empirical experiments show that our model can always outperform the state-of-the-art MSP methods in terms of both accuracy and efficiency.

ACKNOWLEDGEMENTS

This work was supported by NSFC Grant No. 62206067, HKUST-HKUST(GZ) 20 for 20 Cross-campus Collaborative Research Scheme C019 and Guangzhou-HKUST(GZ) Joint Funding Scheme 2023A03J0673, in part by grants from the Research Grant Council of the Hong Kong Special Administrative Region, China (No. CUHK 14217622).

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

## A IMPLEMENTATIONS

### A.1 DATA PREPARATIONS

**Preparing Assembly Graphs.** We start from the residue sequences of the given $N$ chains to form a multimer. We denote the $j$-th residue of the $i$-th chain as $s_j^i$. The embedding function proposed in Chen et al. (2019) produces initial embedding for each residue, denoted as a vector $E(s_j^i)$. Specifically, the embedding vector is a concatenation of two sub-embeddings, which measure the residue co-occurrence similarity and chemical properties, respectively. We average all residue embedding vectors of each chain to obtain the chain-level embedding vectors, i.e., $c_i = \frac{1}{n_i} \sum_j E(s_j^i)$, $\forall 1 \leq i \leq N$, where $n_i$ denotes the residue number of the $i$-th chain. As for a specific multimer, we create the assembly graph $\mathcal{G}_{sou} = (\mathcal{V}_{sou}, \mathcal{E}_{sou})$ whose node attribute represents the pre-computed chain-level embeddings $\{c_i\}_{1 \leq i \leq N}$. Subsequently, according to Algorithm 1 we randomly generated the edge set for the multiemrs. In short, we randomly generate several UCA graphs based on the number of nodes (chains).

---

**Algorithm 1** Formation of $\mathcal{E}_{sou}$.

> Initialization: $p_1 \leftarrow Sample(N, 1), \mathcal{N} \leftarrow \{i \in \mathbb{N}^+ | i \leq N, i \neq p_1\}, \mathcal{Y} \leftarrow p_1, k \leftarrow 1, \mathcal{E} \leftarrow \emptyset$
> **while** $\mathcal{N} \neq \emptyset$ **do**
>     Select $q_k$ from $\mathcal{N}$
>     $\mathcal{N} \leftarrow \mathcal{N} \setminus \{q_k\}$
>     $\mathcal{Y} \leftarrow \mathcal{Y} \cup \{q_k\}$
>     $\mathcal{E}_{sou} \leftarrow \mathcal{E}_{sou} \cup \{(p_k, q_k)\}$
>     $k \leftarrow k + 1$
>     Select $p_k$ from $\mathcal{Y}$
> **end while**

---

**Preparing the Source Task Labels.** We denote the 3D unbound (undocked) structures of $N$ chains to form a multimer as $\{X_i\}_{1 \leq i \leq N}$. In advance, we prepare the set of dimer structures $\{(X_a^{ab}, X_b^{ab})\}_{1 \leq a \leq N, 1 \leq b \leq N}$. For an input assembly graph with the edge index set $\{(e_i^{(1)}, e_i^{(2)})\}_{1 \leq i \leq N-1}$, we follow the Algorithm 2 to obtain the corresponding label.

---

**Algorithm 2** Calculation of $y_{sou}$.

> Initialization: $X'_{e^{(1)}} \leftarrow X_{e^{(1)}}$
> **for** $(e^{(1)}, e^{(2)})$ in $\mathcal{E}_{sou}$ **do**
>     Calculate transformation: $\mathbb{T} \leftarrow \text{Kabsch}(X'_{e^{(1)}}, X_{e^{(1)}}^{e^{(1)}e^{(2)}})$
>     Apply $\mathbb{T}$: $X'_{e^{(2)}} = \mathbb{T}(X_{e^{(2)}}^{e^{(1)}e^{(2)}})$
> **end for**
> **Output:** TM-Score($\{X_i\}_{1 \leq i \leq N}, \{X_i\}'_{1 \leq i \leq N}$)

---

**Preparing Target Data.** For an $N$-chain multiemr, the data for the target task consists of correctly assembled graphs with less than $N$ nodes and one of the remaining nodes. For convenience, we randomly generated multiple assembly graphs with less than $N$ nodes and kept those labeled as 1.0. For each graph, we randomly add one of the remaining nodes and calculated the new label for assembly correctness, which is the final label for the target task. Algorithm 3 show the process to create target dataset using one multimer with $N$ chains. For each element $(G)$ in the output set $B_s$, two nodes at the ends of the last added edge in $G$ are $v_d$ and $v_u$, respectively. Each element in $Y_s$ represents the label $y_{tar}$. We use the output of Algorithm 3 to prepare each data instance $(\mathcal{G}_{tar}, v_d, v_u, y_{tar})$ in $\mathcal{D}_{tar}$.

---

**Algorithm 3** Preparation for $\mathcal{D}_{tar}$.

---

Initialization: $\mathcal{D}_{tar} \leftarrow \emptyset$, $S_u \leftarrow \{1, 2, 3, ..., N\}$ (undocked chain set), $S_d \leftarrow \emptyset$ (docked chain set), $Start \leftarrow Sample(N, 1)$ (starting chain), $B_s \leftarrow \emptyset$ (set of best assembly graphs), $Y_s \leftarrow \emptyset$ (set of target labels)
$B_s \leftarrow B_s \cup \{v_{start}\}$
$S_d \leftarrow S_d \cup \{Start\}$
**for** $i = 1 \rightarrow N - 2$ **do**
   **for** $G$ in $B_s$ **do**
      Update $S_u$ and $S_d$
      **for** $v_d$ in $S_d$ **do**
         **for** $v_u$ in $S_u$ **do**
            Calculate the TM-Score $y$ with Algorithm 2
            **if** $y > 0.99$ **then**
               $G \leftarrow G \cup \{v_u, e_{ud}\}$
               $B_s \leftarrow B_s \cup \{G\}$
               $Y_s \leftarrow Y_s \cup \{y\}$
            **end if**
         **end for**
      **end for**
   **end for**
**end for**
**for** $G$ in $B_s$ **do**
   Update $S_u$ and $S_d$
   **for** $v_d$ in $S_d$ **do**
      **for** $v_u$ in $S_u$ **do**
         Calculate the TM-Score $y$ with Algorithm 2
         $G \leftarrow G \cup \{v_u, e_{ud}\}$
         $B_s \leftarrow B_s \cup \{G\}$
         $Y_s \leftarrow Y_s \cup \{y\}$
      **end for**
   **end for**
**end for**
**Output:** $B_s$ and $Y_s$

---

## A.2 MODEL ARCHITECTURE

**GNN Model for Pre-Training.** We apply source dataset $\mathcal{D}_{sou}$ to pre-train the graph regression model. We denote $H_i^{(k)}$ as the embedding of node $i$ after the $k$-th GIN layer. Therefore, we have the following output with each layer in the GIN encoder.

$$H_i^{(k)} = \text{MLP}_{(k)} \left( (1 + \epsilon^{(k)}) \cdot H_i^{(k-1)} + \sum_{u \in \mathcal{N}(i)} H_u^{(k-1)} \right) \tag{8}$$

where $\epsilon^{(k)}$ represents the learnable parameter of the $k$-th layer.

Finally, we have the GIN encoder output with a 'sum' graph-level readout for the last layer:

$$H = \text{READOUT} \left( \{H_i^{(L)} | 1 \leq i \leq N\} \right) \tag{9}$$

where MLP denotes the Multilayer Perceptron and $L$ means the total layer number.

**Prompt Model.** For a data instance $(\mathcal{G}_{tar}, v_d, v_u, y_{tar}) \in \mathcal{D}_{tar}$ in the target dataset, we consider $v_u$ as an isolated node in graph $\mathcal{G}'_{tar} = (\mathcal{V}_{tar} \cup v_u, \mathcal{E}_{tar})$. The pre-trained GIN encoder computes the node embedding matrix $H$ for $\mathcal{G}'_{tar}$. We obtain the prompt embeddings with a cross attention module:

$$H_x = \sigma_\pi \left( \left[ \text{softmax} \left( H_d^\top H_u \right) H_u^\top \right]^\top \right), \tag{10}$$

$$H_y = \sigma_\pi \left( \left[ \text{softmax} \left( H_u^\top H_d \right) H_d^\top \right]^\top \right), \tag{11}$$

where $\sigma(\pi)$ is a parametric function (3-layer MLP).

## A.3 HYPERPARAMETERS

The choice of hyperparameters of our model is shown in Table 4.

Table 4: Hyperparameter choices of PROMPTMSP.

| Hyperparameters | Values |
|---|---|
| Embedding function dimension (input) | 13 |
| GIN layer number $K$ | 2 |
| Dimension of MLP in Eq. 8 | 1024, 1024 |
| Dimension of $\phi$ in Eq. 1 | 256, 1 |
| Dropout rate | 0.2 |
| Number of attention head | 4 |
| Source/target batch-size | 512, 512 |
| Source/target learning rates | 0.01, 0.001 |
| Task head layer number | 2 |
| Task head dimension | 256, 1 |
| Optimizer | Adam |

## B DATASET

The overall statistic of our dataset PDB-M is shown in Table 5. Overall, we have obtained 9,254 non-redundant multimers after processing and filtering.

- Download all of the multimer structures as their first assembly version
- Remove multimers whose resolution of NMR structures less than 3.0
- Remove the chains whose residue number is less than 50
- If more than 30% of the chains have already been removed from the multiemr, the entire multiemr will be removed.
- Remove all nucleic acids
- Cluster all individual chains on 40% identity with CD-HIT (https://github.com/weizhongli/cdhit)
- Remove the multimer if all of its chains have overlap with any other multimer (remove the subcomponents of larger multimers)
- Randomly select multimers to form the test set and the remaining multimers for training and validation.

Kindly note that due to the generally lower efficiency of the baseline, the size of the test set we divided was relatively small. Moreover, we show the experimental results with a data split according to release date in the next section.

Table 5: Statistics of PDB-M

| $N$ | Train | Valid | Test |
|---|---|---|---|
| 3 | 1325 | 265 | 10 |
| 4 | 942 | 188 | 10 |
| 5 | 981 | 196 | 10 |
| 6-10 | 3647 | 730 | 50 |
| 11-15 | 267 | 53 | 25 |
| 16-20 | 198 | 40 | 25 |
| 21-25 | 135 | 27 | 25 |
| 26-30 | 66 | 14 | 25 |
| **Total** | **7561** | **1513** | **180** |

Table 6: Dataset split based on the released date

| Date | Before (train) | After (test) |
|---|---|---|
| 2000-1-1 | 459 | 8786 |
| 2004-1-1 | 1056 | 8198 |
| 2008-1-1 | 2091 | 7163 |
| 2012-1-1 | 3665 | 5589 |
| 2016-1-1 | 4780 | 4474 |
| 2020-1-1 | 7002 | 2252 |
| 2024-1-1 | 9454 | - |

# C ADDITIONAL EXPERIMENTAL RESULTS

## C.1 DATA SPLIT WITH RELEASE DATE.

We show the results with 6 thresholds of release dates. Using them, we have 6 types of data split based on the entire PDB-M. The data split statistic is shown in Table 6. As the datasets all contain large-scale multimers, we show the comparison only between our method and MoLPC in Table 7.

Table 7: Model performance under the data split based on released dates ($3 \leq N \leq 30$). Threshold represents the boundary separating the training (before) and test (after) sets.

| Date Threshold | 2000-1-1 | 2004-1-1 | 2008-1-1 | 2012-1-1 | 2016-1-1 | 2020-1-1 | Avg. |
|---|---|---|---|---|---|---|---|
| Metric | | | TM-Score (mean) / TM-Score (median) | | | | |
| **Ours(GT)** | 0.27 / 0.24 | 0.42 / 0.35 | 0.42 / 0.42 | 0.47 / 0.50 | 0.52 / 0.49 | 0.57 / 0.54 | 0.45 / 0.42 |
| **Ours(ESMFold)** | 0.31 / 0.28 | 0.33 / 0.29 | 0.34 / 0.36 | 0.36 / 0.36 | 0.38 / 0.38 | 0.37 / 0.41 | 0.35 / 0.35 |

## C.2 RUNNING TIME OF PROMPTMSP

We provide more docking path inference results of our method in Figure 9. We can find that as the scale increases, the inference time for a single assembly process (the orange curve) of our method does not increase, which suggests that the applicability of our model is not limited by the scale.

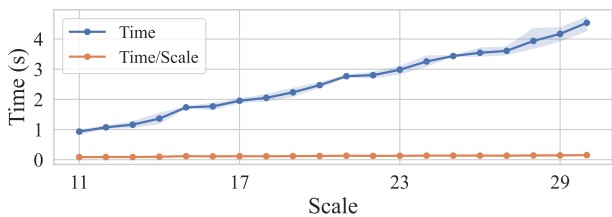

Figure 9: Inference running time of our method tested on various scales of multimers.

## C.3 THE ROLE OF OUR META-LEARNING FRAMEWORK

We test the performance of our method in extreme data scarcity scenarios. In Table 8, data ratio means the proportion of randomly retained multimer samples with chain numbers greater than 10. For example, 10% suggests that we only use 10% of the large-scale multimer data in PDB-M for training. It can be seen that the performance of our model decreases with the degree of data scarcity. However, even with only 10% of the training data retained, our method can still slightly outperform MoLPC. This imples that our method can effectively generalize knowledge from data with fewer chains, without a strong reliance on the amount of large-scale multimer data.

| Data ratio | 80% | 60% | 40% | 20% | 10% |
|---|---|---|---|---|---|
| Metric | | TM-Score(mean) / TM-Score(median) | | | |
| MoLPC | | | 0.47 / 0.45 | | |
| **PromptMSP** | 0.57 / 0.60 | 0.55 / 0.53 | 0.58 / 0.55 | 0.53 / 0.53 | 0.49 / 0.47 |

Table 8: Performance with less training samples.

## C.4  VISUALIZATION

In Figure 10, we demonstrate that PromptMSP can successfully assemble unknown multimers, where no chain has a similarity higher than 40% with any chain in the training set.

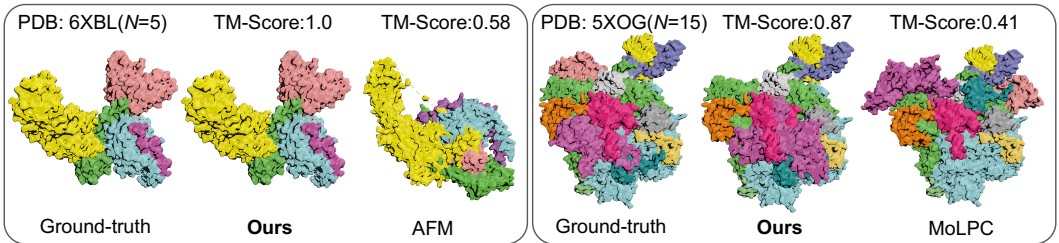

Figure 10: Visualization of multimers with chain numbers of 5 and 15. They are both successfully predicted by PROMPTMSP. For 5XOG, our model correctly predicted 12 out of 14 assembly actions.

## C.5  PROMPT TUNING WITH META-LEARNING

Inspired by the ability of meta-learning to learn an initialization on **sufficient data** and achieve fast adaptation on **scarce data**, we follow the framework of MAML (Finn et al., 2017) to enhance the prompt tuning process. Specifically, we use small-scale multimers (sufficient data) to obtain a reliable initialization, which is then effectively adapted to large-scale multimers (scarce data).

Following Def. 3, we construct datasets $\mathcal{D}_{tar}^{(sma)}$ and $\mathcal{D}_{tar}^{(lar)}$ using data of small-scale ($N \leq 7$) and large-scale multimers ($N \geq 8$), respectively. Let $f_{\pi|\theta^*,\phi^*}$ be the *pipeline* with prompt model ($\pi$), fixed GIN model ($\theta^*$) and fixed task head ($\phi^*$). In our proposed meta-learning framework, we perform **prompt initialization** and **prompt adaptation** using $\mathcal{D}_{tar}^{(sma)}$ and $\mathcal{D}_{tar}^{(lar)}$, resulting in two *pipeline* versions, $f_{\tilde{\pi}|\theta^*,\phi^*}$ and $f_{\pi^*|\theta^*,\phi^*}$ respectively.

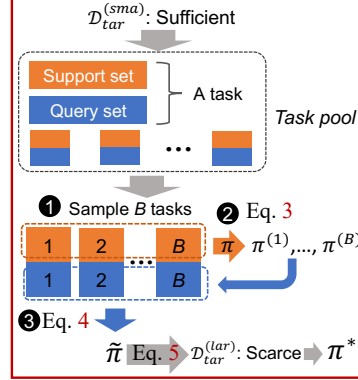

Figure 11: Prompting with MAML.

**Prompt Initialization to obtain $f_{\tilde{\pi}|\theta^*,\phi^*}$.** The objective of prompt initialization is to learn a proper initialization of *pipeline* parameters such that $f_{\tilde{\pi}|\theta^*,\phi^*}$ can effectively learn the common knowledge of $\mathcal{D}_{tar}^{(sma)}$ and performs well on $\mathcal{D}_{tar}^{(sma)}$. Before training, we first create a pool of tasks, each of which is randomly sampled from the data points of $\mathcal{D}_{tar}^{(sma)}$.

During each training epoch, we do three things in order. ❶ We draw a batch of $B$ tasks $\{\mathcal{T}_1, ..., \mathcal{T}_B\}$. Each task $\mathcal{T}_i$ contains a support set $\mathcal{D}_{\mathcal{T}_i}^s$, and a query set $\mathcal{D}_{\mathcal{T}_i}^q$.

❷ We perform gradient computation and update for $\pi$ separately on the support sets of $B$ tasks.

$$\pi^{(t)} = \pi - \alpha \nabla_\pi \mathcal{L}_{\mathcal{D}_{\mathcal{T}_t}^s}(f_{\pi|\theta^*,\phi^*}), \tag{12}$$

where $\pi^{(t)}$ is $\pi$ after gradient update for task $\mathcal{T}_t$.

❸ After obtaining $B$ updated prompt models $\pi^{(t)}, t = 1, 2, ..., B$, the update of $\pi$ for this epoch is:

$$\pi = \pi - \eta \nabla_\pi \sum_{t=1}^{B} \mathcal{L}_{\mathcal{D}_{\mathcal{T}_t}^q}(f_{\pi^{(t)}|\theta^*,\phi^*}). \tag{13}$$

After multiple epochs in a loop (❶, ❷ and ❸ in order), we obtain the prompt model initialization $\tilde{\pi}$.

**Prompt adaptation to obtain** $f_{\pi^*|\theta^*,\phi^*}$ . We apply all data points from $\mathcal{D}_{tar}^{(lar)}$ to update $\pi$ with the prompt initialization $\tilde{\pi}$:

$$\pi^* = \pi - \alpha \nabla_\pi \mathcal{L}_{\mathcal{D}_{tar}^{(lar)}}(f_{\tilde{\pi}|\theta^*,\phi^*}). \tag{14}$$

With Eq. 14, we obtain the prompt adaptation result $\pi^*$.

**Inference under the MAML strategy.** With prompt tuning enhanced by the meta-learning technique, we obtain $\tilde{\pi}$ and $\pi^*$ based on small- and large-scale (chain number) multimers, respectively. For inference on a multimer of $3 \leq N \leq 7$, we perform $N - 1$ assembly steps, in each of which we apply the *pipeline* $f_{\tilde{\pi}|\theta^*,\phi^*}$ to predict the linking probabilities of all pairs of chains and select the most likely pair for assembly. For inference on a multimer of $N \geq 8$ (shown in Figure 5C), we first apply $f_{\tilde{\pi}|\theta^*,\phi^*}$ to assemble a part of the 7 chains of the multimer, and then use $f_{\pi^*|\theta^*,\phi^*}$ for the subsequent $N - 7$ assembly steps.

