# OpenReview forum: "Protein Multimer Structure Prediction via Prompt Learning"
_ICLR.cc/2024/Conference — ICLR 2024 poster_

### Official Review · Reviewer_y71p · 2023-10-18

**Soundness:** 3 good
**Presentation:** 4 excellent
**Contribution:** 3 good
**Rating:** 8
**Confidence:** 3

**Summary:**

The authors introduce an interesting task of multimer structure prediction in the form of assembly graph where each node is a monomer and each edge represents an assembly action. They propose to first pretrain a model to predict the TM-score between the structure obtained from the given assembly graph and the ground truth, then finetune with prompt and meta-learning to perform link prediction to construct the assembly graph step by step. The prompt is crafted with $l=3$ path to form a 4-node graph so that the link prediction can be implemented as graph-level prediction on small graphs which is well aligned with the pretraining phase.

**Strengths:**

1. The paper is well written and clear. I really enjoy reading the paper.
2. The paper introduces an interesting task (i.e. multimer structure prediction) to the community with clear formalization (i.e. prediction of the assembly graph given pairwise dimers).
3. The experiments are solid, testing the performance on multimers ranging from 3 chains to 30 chains. The authors also compare the results when given ground-truth dimers or alphafold-predicted dimers as inputs. The results are promising, exhibiting obvious improvement over baselines.

**Weaknesses:**

1. The $l=3$ graph prompt is proposed to tackle the distribution shift of chain numbers. However, I notice that in section 4.2 the initial embeddings are obtained from the last layer of the pretrained GIN encoder with the full assembly graph as input. This step may already suffer from the distribution shift and produces out-of-distribution embeddings.
2. The ablation of the pretraining phase is missing. An experiment without pretraining should be conducted to demonstrate the necessity of the proposed pretraining strategy.

**Questions:**

1. Can you show the correlation between the number of chains and the node degrees to directly validate the claim "multimers with more chains are more likely to yield assembly graphs with high degrees" in section 3.3?
2. How is the ablation of the C-PPI modelling strategy implemented?

---

> ### Author Response · Authors · 2023-11-21
> **Response to the Reviewer y71p (1/2)**
>
> We sincerely appreciate the reviewer **y71p**'s recognition of our paper and his/her valuable time. We will respond to reviewer's insightful suggestions point-by-point.
>
> **[Cons. 1 GIN encoder distribution shift]** We thank the reviewer for the valuable comments.
>
> The comment relates to the core motivation of this paper. To avoid distribution shift caused by multimers of different scales (chain numbers) in the target task of link prediction, we re-model the link prediction task of **any scale** as a graph-level task for an $l=3$ path (graph). Please kindly note that the node number of such a path is fixed at **4**.
>
> To also avoid distribution shift in the source task (graph-level regression), we should ensure that the scale $N$ of the source data for pre-training the GNN model is around **4**. We mentioned this operation (in our original manuscript) at the end of the section "Reformulating the Target Link Prediction Task" as **"the pre-trained model should always be fed with graphs of similar scales (i.e., node number ranges from 2 to 7) to the source data."**
>
> In fact, to aviod distribution shift on the GIN encoder, we have tried **(1)** only using multimers with $N=4$ for pre-training; **(2)** using sufficient multimer data of $2≤N≤7$ for pre-training. Empirically, we found that the latter (option) led to significantly better model performance. This might be due to the presence of not only gaps but also common knowledge that can be utilized between different scales. Thus, selecting multimer data close to the $N=4$ scale is the optimal choice.
>
> We would like to thank the reviewer once again for the insightful comments, and we will provide an analysis of this issue in the revised manuscript.
>
> **[Cons. 2 Ablation of the pre-training]** Thanks for the valuable suggestion. To address the reviewer's concerns, we establish the baseline without pre-training in the following experiments.
> 1. **Ours:** Our complete version of the PromptMSP model.
> 2. **Ours w/o pt:** Our method without pre-training the GIN encoder $\theta$ and the task head $\phi$.
>
> Specifically, the prompt model still exists in **Ours w/o pt**. Unlike **Ours**, the GIN encoder $\theta$ and the task head $\phi$ are initialized on the target task before training and their parameters are then updated together with the prompt model $\pi$.
>
> The experimental results **(with AFM-produced dimers)** are as follows:
>
> | Chain number $N$ |  3   |  4   |  5   | 6 | 7 | 8 | 9 | 10 |Overall|
> | -------------- |:----:|:----:|:----:|:----:|:-----:|:-----:|:-----:|:-----:|:-----:|
> | Ours w/o pt | 0.64 | 0.60 | 0.45 | 0.38  | 0.40  | 0.33  | 0.34  | 0.34 |0.43|
> | Ours  | 0.71 | 0.66 | 0.54 | 0.50  | 0.46  | 0.46  | 0.45  | 0.40 |0.51|
>
> It can be seen that pre-training the graph-level regression model has a significant effect on almost all chain numbers and average performance.

---

> > ### Author Response · Authors · 2023-11-21
> > **Response to the Reviewer y71p (2/2)**
> >
> > **[Q1 Correlation between $N$ and node degrees]** Thanks for the valuable question. To address your concern, we provide the statistics of our source and target data.
> >
> > First of all, we respectfully explain the definition of **degree** in the original manuscript.
> > * Degree for the source data: the maximum node degree of each assembly graph in the source data
> > * Degree for the target data: the degree of the chain (node) to be docked
> >
> > In the table below, we show the degree proportions of the generated datapoints with multimers of a specific scale. For example, the value in column "degree=2" of row "$N=5$" represents the proportion of datapoints with degree=2 obtained from 5-chain multimers.
> >
> > On the source data:
> > | Proportion |  degree=1   |  degree=2   |  degree=3   | degree=4 | degree=5 |
> > | -------------- |:----:|:----:|:----:|:----:|:-----:|
> > | $N=2$  | 100% | 0 | 0 | 0 |  0 |
> > | $N=3$  | 0 | 100% | 0 | 0 |  0 |
> > | $N=4$  | 0 | 75% | 25% | 0 | 0  |
> > | $N=5$  | 0 | 48% | 48% | 4% | 0  |
> > | $N=6$  | 0 | 58.35% | 39.35% | 1.85% | 0.46%  |
> >
> > On the target data:
> > | Proportion |  degree=1   |  degree=2   |  degree=3   | degree=4 | degree=5 |
> > | -------------- |:----:|:----:|:----:|:----:|:-----:|
> > | $N=3$ | 100% | 0 | 0 | 0 | 0  |
> > | $N=4$  | 67% | 33% | 0 | 0 |  0 |
> > | $N=5$  | 52% | 34% | 12% | 2% | 0  |
> > | $N=6$  | 43% | 27% | 23% | 6% | 1%  |
> >
> > **[Q2 C-PPI ablation]** Thanks for the valuable question.
> >
> > To perform ablation study on C-PPI, we constructed an additional baseline. We still use the proposed pre-training strategy to train the GNN model on assembly graph (source) data. The difference is that we create a new target dataset, where each datapoint contains only two separate chains without any assembled condition. Labels for this target dataset are determined by the presence of physical contact between two chains in the ground-truth multimer structure.
> >
> > For inference, we first randomly choose a chain as the beginning. Then, the entire model can predict the I-PPI probability of any pair of chains. For each step, we select the pair with the highest probability to assemble. After $N-1$ steps, we obtain the $N$-chain multimer assembled by the I-PPI prediction framework.

---

> > > ### Comment · Reviewer_y71p · 2023-11-22
> > > **Thanks for the response**
> > >
> > > Thank you for the detailed response, which largely alleviates my concerns. I will keep my recommendation for acceptance.

---

> > > > ### Author Response · Authors · 2023-11-23
> > > > **Thanks to the reviewer**
> > > >
> > > > Thank you for your positive feedback. We sincerely appreciate your valuable suggestions and the discussions.

---

### Official Review · Reviewer_UpSS · 2023-10-24

**Soundness:** 2 fair
**Presentation:** 1 poor
**Contribution:** 2 fair
**Rating:** 5
**Confidence:** 3

**Summary:**

The paper treats the problem of multimer assembly: given a set of sequences, and the structure of all possible dimers (e.g. from AF2), we wish to assemble the multimer by iteratively selecting the next chain and aligning dimer structures—represented with an assembly graph. The paper proposes a multi-stage solution to this problem. (1) A GNN is pre-trained to predict the multimer TMScore from an assembly graph (2) The “next link” prediction problem is  framed as a TMScore prediction over a fictitious assembly graph, i.e., akin to “prompting” the pretrained GNN. This fictitious assembly graph is created by a “prompting model” and its design is inspired by network-based PPI prediction in bioinformatics. (3) The prompting model–which is specific for each multimer size–is obtained via meta-learning, where the meta-training tasks are small multimer sizes, and the meta-tuning tasks are large multimer sizes.

**Strengths:**

* The paper proposes a novel solution to the difficult problem of multimer structure prediction. Multiple strategies are employed to make this extremely data-scarce problem tractable for deep learning. These strategies are impressive in their sophistication and the bar for originality / novelty has clearly been surpassed.
* The experimental results are good in terms of both performance and runtime relative to the best existing methods.
* The paper is a nice illustration of the concept of learning on top of foundation models such as AF2, a paradigm which arose in NLP and is becoming increasingly useful in biological ML.

**Weaknesses:**

* The paper integrates multiple technical ideas with a complex problem domain, but unfortunately the presentation is very confusing.
   * The paper relies on many ideas that are less familiar to the average reader in protein ML. There should be an extensive background section explaining meta-learning, prompt learning, L=3 PPI prediction, etc.
    * For a procedure with this many moving parts, it is absolutely essential to provide an explicit inference algorithm somewhere.
    * The paper is made even more confusing by certain particular choices of emphasis which serves only to distract the reader on a first pass.
        * It is not clear why it is important to emphasize C-PPI vs I-PPI. Perhaps the authors are trying to draw a distinction with MCTS, but this is really not necessary or within scope. Fully appreciating the difference would require a detailed explanation of the MCTS method, which the paper has no time (or need) to fully explain.
       * The extended discussion in Section 3.3 seems disconnected from the context of the paper and serves only to make it more confusing.
       * The authors repeatedly distinguish between oligomers and multimers based on size, which is very unconventional and should be fixed.
    * L=3 PPI prediction is not obvious and is very confusing when referred to in-passing the first few times it is brought up.

* The pipeline seems unnecessarily complicated and poorly justified. All else held equal, solutions to hard problems should be as simple as possible, and complexity (even if novel) should at least be sensible and easy to justify once understood. Here, it is really not clear why the problem requires such a complex formulation. The so-called source task is a nice way of framing the multimer assembly problem to make it much more data-rich. But then, the most natural solution would seem to be to run the TMScore predictor on all possible next-link additions to the current assembly graph. It seems quite convoluted to instead obtain a prompting model to convert each possible next-link prediction to a fictitious 4-graph when a real (N+1)-graph would also seem to work.

Justification for score. Although I like the wealth of ideas presented in the paper, the presentation is too unclear and the complexity insufficiently justified to recommend acceptance in its current form.

**Questions:**

* Can the authors confirm that there is only one pretrained model, despite the discussion in section 3.3?
* Where are the node embeddings H is used in prompt model? Are only $H_u, H_d$ used?
* Is there precedent for learning a prompt _model_ that generates a different prompt for each input, as opposed to simply learning a _prompt_?
* Are the encoder parameters $\theta$ and task head parameters $\phi$ ever separated? If not, then denoting them separately only makes the paper more confusing.
* How is runtime calculated? I assume the dimer structures are completed "lazily." What explains the large gap in runtime relative to MCTS? It would be nice to report the total number of dimer structures "required" by MCTS vs the proposed method.

---

> ### Author Response · Authors · 2023-11-21
> **Response to the Reviewer UpSS (1/3)**
>
> We sincerely appreciate the reviewer's insightful suggestions on our paper and his/her valuable time. We sincerely appreciate the reviewer's recognition for the contribution and novelty of this paper, and we apologize for any confusion caused by our presentation.
>
> **[Cons. 1. Background introduction]** Thank you for bringing this important issue to our attention. It is necessary to provide an introduction to the background knowledge involved. Therefore, we will add relevant explanations in the revised manuscript.
>
> For clarity, we briefly introduce the background knowledge here.
> * Meta-learning is a technique that aims to enable models to quickly and efficiently learn from and adapt to new data. Meta-learning is a training technology rather than a deep neural network. **It is well-known for its ability to learn common knowledge from sufficient data and quickly adapt to (new) scarce data.** For the multimer structure prediction (MSP) task, there is a severe imbalance in the number of scales (chain numbers) among multimers, which can be intuitively well-addressed by meta-learning techniques.
> * In Section 2, we have introduced the background of prompt learning. In short, prompt learning can narrow the gaps between different data or tasks by **introducing learnable flexibility to the model without the need for fine-tuning the pre-trained model.** For the MSP task, we verified that regardless of whether the problem is modeled as TM-Score prediction or link prediction, there is a knowledge gap caused by the difference in chain numbers (scales) among multimers. This challenge leads to poor performance of training a unified model using data from all scales (please kindly refer to the experimental results in **[Cons. 5]**). **Motivated by this, we aim to use prompt learning to establish connections between data of multiple scales, which allows us to train a unified (prompt) model to handle all samples.**
> * **$l=3$ PPI path:** We apologize for the missing explanation. We respectfully explain that $l=3$ path is an important identifier for PPI prediction. The existence of such a path implies that the proteins at both ends of the path will undergo PPI. Since $l=3$ path always has only 4 nodes, this allows us to re-model the link prediction task for multimers of different scales as a 4-node graph-level regression task (TM-Score prediction).
>
> We sincerely thank the reviewers once again for bringing to our attention the importance of these key background knowledge.
>
> **[Cons. 2. Inference process]** Thanks for the valuable comment. We have reorganized Figure 5 (framework diagram) and added the description of the inference process (Section 4.4).
>
> The inference process is multi-step. For an $N$-chain multimer, we randomly choose a starting chain and then perform $N-1$ assembly steps in order. During each step, we predict linking probabilities of all possible chain pairs and then select the most possible pair for assembly.
>
> **[Cons. 3. C-PPI, Section 3.3 and scales]** We thank for the valuable suggestions. We respectfully merge these three questions together because they are closely connected.
>
> One of the purposes of emphasizing C-PPI is indeed to distinguish our method from MCTS and to demonstrate that using only I-PPI modeling for MSP tasks is biased. However, another crucial factor to consider is that predicting C-PPI (i.e. link prediction) with different chain numbers of multimers can lead to the out-of-distribution (OOD) situation for model's generalization. For clarity, we use "10+1" to denote predicting the next right link on an already assembled 10-chain multimer. **Such OOD issue means that we struggle to use a model trained on "10+1" (C-PPI) data to perform the "29+1" prediction.** This is why we emphasize the importance of "scale" in Section 3.3 and throughout the entire paper.
>
> We apologize for any confusion caused by our unclear presentations. We will remove the wording "oligomers" and restate the relationship between C-PPI and scale in the introduction part.

---

> > ### Author Response · Authors · 2023-11-21
> > **Response to the Reviewer UpSS (2/3)**
> >
> > **[Cons. 4. $l=3$ path]** We thank the reviewer for the valuable comments. We will supplement a brief description in the introduction.
> >
> > In addition to its special biological interpretation, the $l=3$ path plays a practical role in unifying varied scales in this paper. **This path (graph) allows us to transform any scale of link prediction task into a 4(fixed)-node graph-level TM-Score prediction paradigm.** For example, if we want to predict a specific linking probability between an undocked chain "u" and a chain "d" in an already assembled 10-chain multimer (i.e., "10+1" case), we obtain the embeddings of "u" and "d" directly from our pre-trained GNN model and compute the embeddings of the other two virtual nodes "x" and "y". Embeddings of "u", "d", "x" and "y" jointly form the $l=3$ path. **As a result, the "10+1" prediction task is successfully transformed into a 4-node TM-Score prediction task.** Kindly note that, regardless of whether it is a "10+1" or a "20+1" case, the linking probability is always obtained from a path (graph) with 4 nodes.
> >
> > **[Cons. 5. Complex task formulation]** We sincerely appreciate the reviewer for the insightful comment.
> >
> > We respectfully clarify that the most intuitive solution for the MSP task is, as the reviewer mentioned, to use only a TM-Score predictor to gradually select the optimal assembly state for each step. This was also our initial attempt in this work. However, we still have to respectfully emphasize the out-of-distribution (OOD) problem caused by the difference in scales (chain numbers).
> >
> > We would like to present the following confirmatory experiment.
> >
> > Specifically, "**$N$ for training**" means that the scale(s) of the multimers we use to train the TM-Score predictor (i.e., predicting the TM-Score of a given assembly graph). For all $10$ models, our test task is the same, which is the MSP task of multimers with $N=7$.
> >
> > |$N$ for training  |  3   |  4   |  5   | 6 | 7 | 8 | 9 | 10 |6+7|3,4,...,30|
> > | -------------- |:----:|:----:|:----:|:----:|:-----:|:-----:|:-----:|:-----:|:-----:|:-----:|
> > | Performance | 0.32 | 0.35 | 0.33 | 0.38  | 0.40  | 0.36  | 0.29  | 0.26 |0.43| 0.22 |
> >
> > We have several findings:
> > 1. The scale of multimers used for training is recommended to closer to 7. The further the scale $N$ is from 7, the worse the testing performance.
> > 2. Training the model with a combination of multimers with $N=6$ and $N=7$ yields better results than using only $N=7$ multimers for training.
> > 3. Training the model with a combination of multimers of all scales (3, 4, ..., 30) yields worse results than using only $N=7$ multimers for training.
> >
> > These findings confirm the OOD issue caused by the scales mentioned above. There are connections and gaps between multimers of different scales. Thus **training a separate model for each $N$ or training a unified model with all scales together are both not the ideal approaches.**
> >
> > The prompt learning we proposed aims to maximize the utilization of common knowledge between different scales, enabling to apply a unified model to handle all scales well.

---

> > > ### Author Response · Authors · 2023-11-21
> > > **Response to the Reviewer UpSS (3/3)**
> > >
> > > **[Q1 Only one pretrained model?]** Thanks for the insightful question. Experiments in Section 3.3 and **[Cons. 5]** were carried out using **multiple pre-trained models** because we aim to demonstrate the impact of scale differences on the model's performance. However, in our framework, we only need one pre-trained model to handle the inference of multimers of all scales.
> > >
> > > **[Q2 Prompt embeddings]** Yes, only $H_u$ and $H_d$ are used as inputs to the prompt model to compute embeddings ($H_x$ and $H_y$) for virtual nodes.
> > >
> > > In fact, due to the node aggregation operation of the pre-trained GNN model, chain "d" also carries information from other chains. That is, the currently assembled state (neighbor nodes of node "d") will determin the embedding $H_d$.
> > >
> > > **[Q3 Prompt results]** Thanks for the valuable question. The prompt embedding in this paper is calculated by a learnable prompt model. All multimers share the same trained prompt model, but **they get different prompt embeddings**. In short, our model generates different $l=3$ paths for different input samples.
> > >
> > > Research in the field of prompt learning shows that the prompt results may be fixed or different for different inputs. Earlier prompt engineering tended to provide fixed (hand-crafted) prompts for different data, such as GPT-3 **[re1]**. For example, if we want to prompt the language model to predict the speaker's mood when speaking, the prompt can be fixed as `I feel so`. Recent prompt learning research **[re2,re3,re4]** tends to increase the flexibility for the entire model with trainable prompts. In this way, even if the pre-trained model is not fine-tuned, the prompt model can capture the relationship between different sources of data or tasks.
> > >
> > > **[Q4 $\theta$ and $\phi$]** We respectfully clarify that $\theta$ and $\phi$ are separate. We only use the GIN encoder $\theta$ to calculate chain embeddings ($H_u$ and $H_d$). After obtaining the $l=3$ path, $\theta$ and $\phi$ work together to obtain the final linking probability.
> > >
> > > **[Q5 Inference time]** Thanks for the valuable question. Yes, the proposed framework aims to predict accurate docking paths for various types of dimer structures. Therefore, we did not consider the time of dimer structure computations. This is indeed unfair compared to other baselines (except for MoLPC), so we provide additional results to show the running time of both docking path prediction and dimer structure prediction separately. However, we respectfully clarify that the runtime of MoLPC shown in the original manuscript also does not include the time for preparing dimer structures.
> > >
> > > **The name in the parentheses represents applied method for dimer structure prediction.**
> > > | Running time (hour) |  Docking Path   |  Dimer structure   | Total |
> > > | -------------- |:----:|:----:|:----:|
> > > | Multi-LZerD | N/A | N/A | 1.357 |
> > > | RL-MLZerD | N/A | N/A | 1.591 |
> > > | AFM | N/A | N/A | 0.640 |
> > > | MoLPC(AFM) | 0.159 | 1.457 | 1.616 |
> > > | **Ours(AFM)** | 0.003 | 0.559 | 0.562 |
> > > | **Ours(EquiDock)** | 0.003 | 0.007 | 0.011 |
> > > | **Ours(ESMFold)** | 0.003 | 0.002 | 0.005 |
> > > | **Ours(HDock)** | 0.003 | 1.050 | 1.054 |
> > >
> > > **For multimers of $3\leq N \leq 10$ in the test set, MoLPC on average needs to calculate 13.9 dimer structures per multimer, while our method only needs 5.5.** This is because MoLPC needs to try to dock multiple dimer structures during the search process, while ours does not require dimer structures as input for predicting the docking path in the inference phase. For a multimer with $N$ chains, our method needs $N-1$ dimer structures for inference.
> > >
> > > **[re1]** Brown T, Mann B, Ryder N, et al. Language models are few-shot learners. NeurIPS, 2020.
> > >
> > > **[re2]** Sun X, Cheng H, Li J, et al. All in One: Multi-Task Prompting for Graph Neural Networks. SIGKDD, 2023.
> > >
> > > **[re3]** Fang T, Zhang Y M, Yang Y, et al. Universal prompt tuning for graph neural networks. NeurIPS, 2023.
> > >
> > > **[re4]** Li X L, Liang P. Prefix-tuning: Optimizing continuous prompts for generation. ACL, 2021.

---

> > > > ### Author Response · Authors · 2023-11-23
> > > > **A friendly reminder for discussion**
> > > >
> > > > As the discussion period is nearing its end, we would appreciate it if you could confirm whether our responses have addressed your concerns. If you have any additional comments, we are more than willing to engage in further discussion.
> > > >
> > > > We will continue to make efforts to improve the clarity of the paper by modifying the presentation. Moreover, we will release the code and checkpoints to facilitate the reproducibility and further research in this area.
> > > >
> > > > Thank you very much for your time and consideration.

---

### Official Review · Reviewer_6yks · 2023-10-31

**Soundness:** 2 fair
**Presentation:** 1 poor
**Contribution:** 3 good
**Rating:** 5
**Confidence:** 3

**Summary:**

This paper proposes a sequential protein complex assembly method called PromptMSP. In each assembly step, PromptMSP predicts where a protein should be assembled to the current complex. During training, PromptMSP learns a continuous score for a given protein assembly graph and during testing, it uses the learned score model to find the most likely assembly graph. To avoid training and testing distributional mismatch, PromptMSP employs prompt learning to reduce the gap of input formats. PromptMSP is compared with existing multimer prediction baselines and outperforms AlphaFold-multimer baseline.

**Strengths:**

* The proposed method outperforms AlphaFold-Multimer (AFM), which is impressive.
* The evaluation setting is comprehensive. It includes both ground truth dimer setting and predicted dimer setting, which ensures a fair comparison with AFM.
* Ablation studies show that each proposed component is effective.
* Incorporation of L=3 PPI rule into the inference procedure is an interesting contribution.

**Weaknesses:**

* The method description is very confusing. Figure 5 is very crowded and rather uninformative.
* It is very hard to understand what meta-learning part (section 4.3) is actually doing. A visual step-by-step illustration of prompt fine-tuning can be helpful.
* The introduction of prompt fine-tuning seems an overkill. A simpler approach should work equally well. For example, we can adopt a standard autoregressive link prediction algorithm to this problem. In each step, you predict the link between a pair of proteins and train the model to predict the right link given different prefix graphs.
* Analysis in section 3.3 is unclear. How did you compute Centered Kernel Alignment between two models?
* It's unclear how a new protein is docked to the current assembly in each step. Did you use EquiDock? If so, how do you ensure that EquiDock is not trained on any of your test set instances?

**Questions:**

* At test time, what prompt do you provide to the model? It seems that the prompt is basically the assembly graph that model predicted. I don't see why prompt engineering is useful during training.
* It would be helpful to report model performance for each number of chain (from 3 to 30).

---

> ### Author Response · Authors · 2023-11-21
> **Response to the Reviewer 6yks (1/3)**
>
> We thank the reviewer **6yks** for acknowledging our method as "impressive", our experiments as "comprehensive" and our prompt method as "interesting contribution". Many thanks for the constructive feedback especially for the questions about the presentation and our detailed settings. **These all help us to further enhance the readability and the quality of our paper.**
>
> To address reviewer 6yks’s concerns, we provide point-wise responses below.
>
> **[Cons 1. Method description and Figure 5's organization]** We sincerely apologize for the confusion caused. To address this issue, we have **(1) clearly described the proposed method and (2) made revisions to Figure 5 and the description of meta-learning part. We sincerely appreciate your suggestions, as they have indeed improved the readability and quality of our paper.**
>
> 1. Descriptions:
>
> 1.1 **The motivation of our paper:** Under the multimer structure prediction (MSP) task, our main goal is to address the impact of knowledge gap of multimer scales (chain numbers) on C-PPI prediction.
>
> 1.2 **C-PPI knowledge gaps:** In Section 3.3, we validated the existence of gaps and connections in C-PPI knowledge implied by multimers of different scales (chain numbers).
>
> 1.3 **$\mathscr{l}=3$ rule prompting (main contribution):** We first **pre-train** the GNN model, which takes an assembly graph as input. Its encoder $\theta^*$ outputs the node-level embeddings and the head $\phi^*$ outputs the predicted TM-Score. During tuning, we update only the prompt model $\pi$ based on the target data (conditional link prediction). For clarity, we refer to the combination of these three modules as a `pipeline`, denoted as $f_{\pi|\theta^*,\phi^*}$. In short, the `pipeline` $f_{\pi|\theta^*,\phi^*}$ outputs the linking probability conditioned on an already assembled portion of the multimer and a newly added chain.
>
> 1.4 **Meta-learning:** We respectfully clarify that the meta-learning method we apply is **a training strategy**, rather than **a neural network**. It is responsible for reliable prompt tuning, i.e., to update $\pi$ within $f_{\pi|\theta^*,\phi^*}$.
>
> 2. Revisions:
>
> 2.1 **We have re-organized Figure 5,** where we remove the meta-learning part and clearly show the specific process of training (pre-training, prompt-tuning) and testing.
>
> 2.2 We explicitly state in Section 4.3 that meta-learning is **a training strategy** and **explain the purpose** of applying this technique.
>
> 2.3 We have added a schematic diagram (Figure 7) to show the process for obtaining prompt tuning results.
>
> 2.4 We have included a part (Section 4.4) to introduce the inference stages.
>
> **All revisions can be found in our revised manuscript. We sincerely apologize for any confusion we may have caused.**
>
> **[Cons. 2 The meta-learning part's readability]** Thanks for your valuable suggestions. To make the applied meta-learning technique much clearer, we have revised the entire Section 4.3 and **visualized specific cases to illustrate how it is responsible for prompt tuning.**
>
> **In response, we describe the process as follows.**
> * Assuming that the pre-trained model has been well-trained, now we have the GIN encoder $\theta^*$ and the task head $\phi^*$. **They will be frozen from now on.**
> * Our goal of prompt tuning is to obtain 2 different prompt models $\tilde{\pi}$ and $\pi^*$ for scales $3\leq N\leq 7$ and $N>7$, respectively. For example, to infer a $10$-chain multimer, we perform 9 assembly steps. In each of the first 6 steps, we apply the `pipeline` $f_{\tilde{\pi}|\theta^*,\phi^*}$ to predict the
> linking probabilities of all pairs of chains and select the most likely pair for assembly. Upon this assembled $7$-chain multimer, we apply the `pipeline` $f_{\pi^*|\theta^*,\phi^*}$ to sequentially add the remaining 3 chains.
> * **The process of getting $\tilde{\pi}$:** We construct a pool of **tasks** from the link prediction datapoints created from multimers of $3\leq N\leq 7$. Each task contains two sets (a support set and a query set) of link prediction datapoints. **In each epoch, we do three things in order**. (1) Randomly sample $B$ tasks from the task pool; (2) for each of the $B$ support sets, update a separate model starting from $\pi$, resulting in $\pi^{(i)},i-1,...,B$; (3) compute the sum of loss ($\mathcal{L}\_{meta}$) of these all $\pi^{(i)},i=1,...,B$ on the query sets and update $\pi$ with the gradient of $\mathcal{L}_{meta}$ with respect to $\pi$. **After multiple epochs of iteration, we obtain prompt model $\tilde{\pi}$.**
> * **The process of getting** $\pi^*$: We create link prediction datapoints from multimers of $8\leq N\leq 30$. Over multiple epochs, we update the previously obtained $\tilde{\pi}$ to $\pi^*$.

---

> > ### Author Response · Authors · 2023-11-21
> > **Response to the Reviewer 6yks (2/3)**
> >
> > **[Cons. 3 The role of prompt fine-tuning]** Thanks for the valuable suggestions. We fully agree with your description of our task as "predicting the right link given different prefix graphs". Actually, our target task is to follow a similar design, which involves using a prefix graph to predict the linking probability between each undocked chain and each node of the prefix graph. Finally, we take the most likely link as the right link.
> >
> > However, one of the contributions of our paper is to verify the existence of a knowledge gap (Figure 4) in the target task for multimers of different scales (chain numbers). **In short, it difficult to train a unified model that can well handle the assembly of both a 5-chain multimer and a 30-chain multimer.**
> >
> > Inspired by the prompt learning methods, which increases the model's flexibility to narrow the gap between different data or tasks, we adopt a prompt model to construct the "bridge", $\mathscr{l}=3$ path, which can be applied to multimers of varied scales.
> >
> > The empirical results also validate the importance of prompting. Table 3 shows that without the prompt model and only predicting the right link in order, our model's performance would be significantly reduced.
> >
> > **[Cons. 4 Centered Kernel Alignment]** Thanks for the valuable question. We sincerely apologize for not providing a detailed explanation of the calculation process with Centered Kernel Alignment (CKA).
> >
> > We use Figure 4 to show the knowledge gaps between multimers with varied scales (chain numbers). For example, if we want to analyze the relationship between data of 5-chain and 25-chain multimers, we first train separate 2 models for each of them. We then use the testing multimer as input for such two well trained models and obtain its two versions of representations $\text{X}$ and $\text{Y}$.
> >
> > **CKA **[re1]** is widely used to compute the relationship between the representations of two models.** First, we compute the Gram matrices for two representations, $\text{K}=\text{X}\text{X}^{\text{T}}$ and $\text{L}=\text{Y}\text{Y}^{\text{T}}$. Then CKA computes:
> > $\text{CKA}(\text{K},\text{L})=\frac{\text{HSIC}(\text{K},\text{L})}{\sqrt{\text{HSIC}(\text{K},\text{K})\text{HSIC}(\text{L},\text{L})}}$，
> > where $\text{HSIC}$ is the Hilbert-Schmidt independence criterion.
> >
> > We find that the model the similarity between two models is quite low (about 0.2), indicating that there are significant gaps between these 2 kinds of training data (5-chain and 25-chain multimers).
> >
> > **[Cons. 5 Data leakage]** During each step of inference, we predict linking probabilities of all possible chain pairs and then select the most possible pair (right link) to dock. In our experiment part, we apply AlphaFold-Multimer (AFM) to provide dimer structures for each assembly (docking) action.
> >
> > We have ensured that there is no overlap between our test set and AFM's training set. The description of the training set used in the original AFM paper **[re2]** is as follows: "The training dataset consisted of structures from the Protein Data Bank (PDB) with a maximum release date of 2018-04-30."
> >
> > **In our paper, we ensured that:**
> > 1. The release date of the multimers included in our test set is no earlier than 2018-04-30;
> > 2. The sequence similarity between any chain in our test set and any chain in our training set is no more than 40%.
> >
> > These two assurances indicate that **there is no data leakage between our test set and AFM's training set**, and there is also no data leakage between our training set and test set.

---

> > > ### Author Response · Authors · 2023-11-21
> > > **Response to the Reviewer 6yks (3/3)**
> > >
> > > **[Q1 The process of prompting]** Thanks for the question. We respectfully clarify that we do not introduce any additional information for prompting. At test time, the prompt model automatically constructs **a virtual path (graph)**. During pre-training, we have trained an excellent GNN model to evaluate the correctness of any input graph. Therefore, the pre-trained GNN model evaluates the correctness of the virtual graph constructed by the prompt model, i.e., the probability that two chains have an assembly action (link).
> > >
> > > The prompt engineering **[re3,re4]** has been widely proven to narrow the gap between different data (or tasks) by increasing the model's flexibility. In our paper, we focus on the biggest challenge of the multimer structure prediction task: the gaps caused by the varied scales of multimers make it difficult to train a unified model that can be applied to all scales of multimers. **Therefore, prompting technique is naturally suitable for solving this challenge. The unified prompt model is designed to capture the common knowledge of multimers, which enables it to bridge all scales of multimers.**
> > >
> > > **[Q2 Performance for each $N$ (chain number)]**  Thanks for this valuable insights. We strongly agree that reporting the performance of each $N$ for all methods is crucial. To this end, we present the performance (mean TM-Score) of all baselines over different numbers of chains in the table below. **It is evident that as the chain number increases, the performance of all baselines tends to decline.**
> > >
> > > Setting: "AFM Dimer", i.e., both MoLPC and our method apply AFM-produced dimer structures for assembly.
> > > The best performance is **bolded** and the second best is `highlighted`.
> > > | Chain Number | 3 | 4 | 5 | 6 | 7 | 8 | 9 | 10| overall |
> > > |---|---|---|---|---|---|---|---|---|---|
> > > | Multi-LZerD | 0.48 | 0.47 | 0.35 | 0.32 | 0.32 | 0.33 | 0.27 | 0.29 | 0.35 |
> > > | RL-MLZerD | 0.45 | 0.44 | 0.34 | 0.37 | 0.38 | 0.32 | 0.27 | 0.24 | 0.36 |
> > > | AFM | `0.68` | `0.53` | 0.52 | `0.45` | `0.42` | `0.44` | `0.38` | `0.38` | `0.46` |
> > > | MoLPC | 0.62 | `0.53` | **0.56** | 0.43 | 0.36 | 0.37 | 0.30 | 0.28 | 0.42 |
> > > | **Ours** | **0.71** | **0.66** | `0.54` | **0.50** | **0.46** | **0.46** | **0.45** | **0.40** | **0.51** |
> > >
> > > Since only MoLPC and our method can handle multimers with chain numbers greater than 10, the results of only these two methods are reported in the table below.
> > > | Chain Number | 3 | 4 | 5 | 6 | 7 | 8 | 9 | 10| 11 | 12|13|14|15|16|17|18|19|20|21|22|23|24|25|26|27|28|29|30|Overall|
> > > |---|---|---|---|---|---|---|---|---|---|---|---|---|---|---|---|---|---|---|---|---|---|---|---|---|---|---|---|---|---|
> > > | MoLPC| 0.62 | 0.53 | 0.56 | 0.43 | 0.36 | 0.37 | 0.30 | 0.28 | 0.35| 0.36 | 0.33|0.29|0.28|0.33|0.42|0.25|0.26|0.25|0.30|0.28|0.25|0.25|0.23|0.23|0.51|0.29|0.28|0.24|0.38|
> > > | Ours | 0.71 | 0.66 |0.54 | 0.50 | 0.46 | 0.46 | 0.45 | 0.40 | 0.37| 0.49 | 0.41|0.45|0.40|0.36|0.33|0.37|0.37|0.33|0.37|0.35|0.36|0.43|0.32|0.37|0.33|0.33|0.26|0.36|0.46|
> > >
> > > **[re1]** Raghu M, Unterthiner T, Kornblith S, et al. Do Vision Transformers See Like Convolutional Neural Networks? NeurIPS, 2021.
> > >
> > > **[re2]** Evans R, O’Neill M, Pritzel A, et al. Protein complex prediction with AlphaFold-Multimer. biorxiv, 2021.
> > >
> > > **[re3]** Sun X, Cheng H, Li J, et al. All in One: Multi-Task Prompting for Graph Neural Networks. SIGKDD, 2023.
> > >
> > > **[re4]** Fang T, Zhang Y, Yang Y, et al. Prompt tuning for graph neural networks. NeurIPS, 2023.

---

> > > > ### Comment · Reviewer_6yks · 2023-11-21
> > > > **Thank you for your response**
> > > >
> > > > Thank you for your detailed response. Overall, it is still hard to understand the methodology of this paper because it has so many components. Therefore, I would like to keep my original score

---

> ### Author Response · Authors · 2023-11-23
> **Follow up response**
>
> Thank you very much for your feedback. We sincerely believe that we have made every effort to organize the paper and that the revised version has greatly improved its clarity and readability.
>
> Overall, our method includes two parts: the model part and the meta-learning part. The basic definition of meta-learning is **Model-Agnostic**, which is a technique that enables the prompt model to obtain more reliable training. In addition, existing work **[re1]** has already demonstrated that meta-learning can significantly improve the effectiveness of prompt techniques.
>
> If you could provide more specific information on what is unclear, we would be more than happy to continue improving the readability of the paper. Besides, we will release the code and checkpoints to facilitate the reproducibility and further research in this area.
>
> **[re1]** All in One: Multi-Task Prompting for Graph Neural Networks. SIGKDD (best research paper award), 2023.

---

### Official Review · Reviewer_PqHe · 2023-11-09

**Soundness:** 2 fair
**Presentation:** 2 fair
**Contribution:** 2 fair
**Rating:** 3
**Confidence:** 4

**Summary:**

This paper introduces a new algorithm to predict multimer structure with multiple chains via a pre-training and prompt tuning framework. The overall idea is novel and interesting. Different from MoLPC, where proteins docking are independent without the consideration of other protein, this method considers the influence of third-party proteins when performing docking. This paper compared several baselines on N chains datasets (N>=3). The experimental results show improvement on AlphaFold-Multimer and MoLPC. Although this paper introduces some new idea, many details are unclear. Also, the baselines are so weak and the experimental setting is not realistic.
I vote to reject this paper.

**Strengths:**

- Solving multimer structure prediction via pre-training and prompt tuning is interesting.

- It's reasonable to consider conditional docking for multiple protein.

**Weaknesses:**

- defintion 1 is problematic. Because in real-world setting for docking, monomer's ground-truth structures could not be provided. So that the correctness could never be 1 in real-world setting.

- the baselines are so weak. when taking ground-truth structure as input, HDock[2] and xTrimoDock[3] are strong baselines.

- it could be interesting if you can compare different baselines over different the number of chains. The performance could be reduced when increasing the number of chains.

- missing some related references: [1], [3], [4] [5]

[1] Ghani, Usman, et.al. Improved docking of protein models by a combination of alphafold2 and cluspro.

[2] Yumeng Yan, et.al. The HDOCK server for integrated protein–protein docking.

[3] Yujie Luo, et.al. xTrimoDock: Rigid Protein Docking via Cross-Modal Representation Learning and Spectral Algorithm.

[4] Mohamed Amine Ketata et.al. DIFFDOCK-PP: RIGID PROTEIN-PROTEIN DOCKING WITH DIFFUSION MODELS. use torsional diffusions to solve rigid protein docking, and the source code is released.

[5] Lee-Shin Chu et.al. Flexible Protein-Protein Docking with a Multi-Track Iterative Transformer.

**Questions:**

- when comparing with AlphaFold-Multimer, do you input monomer's ground-truth structure as the template?

- how does your method perform when using predicted monomer structure? is the method robust?

---

> ### Author Response · Authors · 2023-11-16
> **Response to the Reviewer PqHe (1/2)**
>
> We thank the reviewer **PqHe** for acknowledging our methodology is “interesting” and our motivation is “reasonable”. We sincerely appreciate the reviewer’s valuable time and comments. We provide point-wise responses below.
>
> **[Cons. 1 Monomer's ground-truth structures]** We acknowledge and apologize for the ambiguity in the task objective stated under Definition 1. We will revise it as: *With the above definitions, our paper aims to predict assembly graphs (>>2 proteins chains) that maximize the TM-Scores, taking as inputs the residue sequences of proteins and pre-calculated dimer structures (=2 protein chains).*
> **However, we respectfully clarify that pre-calculated dimer structures can be obtained from AlphaFold-Multimer without the need for providing ground-truth monomers.**
>
> We strongly agree that the ground-truth monomer or ground-truth dimer structures should not be used in real-world setting. We respectfully emphasize that **our approach did not rely on ground-truth monomer to perform inference in real-world setting**.
>
> |Setting|Inference inputs|Inference intermediate variables |Inference outputs|
> |-|-|-|-|
> |AFM Dimer|Amino acid sequences|Predicted docking path, Dimer structures predicted by AlphaFold-Multimer|Predicted multimer structure|
> |GT Dimer|Amino acid sequences, Ground-truth dimer structures|Predicted docking path|Predicted multimer structure|
>
> Indeed, in this work we propose a framework to predict the best possible protein multimers (>>2 protein chains), taking as inputs the pre-calcuated dimer structures (=2 protein chains). For the pre-calcuated dimer structures, we considered two settings in our **original experiments:**
> * The "AFM Dimer" setting does not involve ground-truth monomer nor ground-truth dimer structures. We predict dimers with AlphaFold-Multimer by inputting only amino acid sequences and our model also only takes amino acid sequences as inputs for predicting the docking path. This setting is applicable to real-world scenarios and consistent with the experimental setup of MoLPC **[re1]** (our baseline).
> * The "GT Dimer" setting may be less realistic under the application scenarios, but we respectfully think that it is not completely meaningless. The performance under this "GT Dimer" setting suggests the effectiveness of the docking paths predicted by ours and the MoLPC baseline.
>
> **[Cons. 2 HDock and xTrimoDock]** We sincerely thank the reviewer's comment. **Firstly, we respectfully clarify once again that our "AFM Dimer" setting does not require any ground-truth monomers as input for real-world inference.**
> * **HDock and xTrimoDock may not be suitable for serving as the baselines.** We have checked HDock's local version and online server, and inquired the HDock team about the potential of performing multimer structure prediction (MSP). We find that HDock is currently unable to perform hetero-protein MSP tasks. We also checked all versions related to xTrimoDock **[re2,re3,re4]**, and even went through the supplementary materials in its NeurIPS'23 submission, but we could not find any available code. **Thus, we could not use HDock and xTrimoDock as baselines to directly predicting multimer structures.**
> * However, we greatly appreciate the suggestion for taking HDock into consideration and have added the "HDock Dimer" setting, where **we use HDock to pre-calculate dimer structures**.
> * To further demonstrate that our method is not dependent on ground-truth monomers or ground-truth dimers, we also added the "ESMFold Dimer" setting, where **we use sequences to predict dimer structures with ESMFold**. Under all three settings, our method consistently and significantly outperforms the MoLPC and AlphaFold-Multimer baselines.
>
> **Overall, we sincerely appreciate the suggestion about HDock which helps yield surprisingly satisfactory experimental results. We consider HDock as a setting to provide pre-calculated dimer structures, rather than a baseline that can directly perform MSP.**
>
> | Setting | AFM Dimer | AFM Dimer | HDock Dimer | HDock Dimer |ESMFold Dimer | ESMFold Dimer |
> |---|---|---|---|---|---|---|
> | Model | Mean TM-Score | Mean RMSD | Mean TM-Score | Mean RMSD | Mean TM-Score | Mean RMSD |
> | AFM | 0.46 | 31.17 | 0.46 | 31.17 | 0.46 | 31.17 |
> | MoLPC | 0.42 | 34.21 | 0.50 | 30.76 | 0.39 | 40.73|
> | **Ours** | **0.51** | **27.77** | **0.54** | **25.39** | **0.49** | **28.34** |
>
> Detailed experimental results will be shown in Table 9 in the Appendix of our revised manuscript.

---

> > ### Author Response · Authors · 2023-11-16
> > **Response to the Reviewer PqHe (2/2)**
> >
> > **[Cons. 3 Results over different chain numbers $N$]** Thank you for your valuable insights. Yes, this is important for validating the effectiveness of the proposed method and better understanding the challenges of the MSP task. To this end, we present the performance (mean TM-Score) of all baselines over different numbers of chains in the table below. **It is evident that as the chain number increases, the performance of all baselines across all three settings tends to decline.**
> >
> > Setting: "AFM Dimer". Best performance is **bolded**.
> > | Chain Number | 3 | 4 | 5 | 6 | 7 | 8 | 9 | 10| overall |
> > |---|---|---|---|---|---|---|---|---|---|
> > | Multi-LZerD | 0.48 | 0.47 | 0.35 | 0.32 | 0.32 | 0.33 | 0.27 | 0.29 | 0.35 |
> > | RL-MLZerD | 0.45 | 0.44 | 0.34 | 0.37 | 0.38 | 0.32 | 0.27 | 0.24 | 0.36 |
> > | AFM | 0.68 | 0.53 | 0.52 | 0.45 | 0.42 | 0.44 | 0.38 | 0.38 | 0.46 |
> > | MoLPC | 0.62 | 0.53 | **0.56** | 0.43 | 0.36 | 0.37 | 0.30 | 0.28 | 0.42 |
> > | **Ours** | **0.71** | **0.66** | 0.54 | **0.50** | **0.46** | **0.46** | **0.45** | **0.40** | **0.51** |
> >
> > Setting: "HDock Dimer"
> > | Chain Number | 3 | 4 | 5 | 6 | 7 | 8 | 9 | 10| overall |
> > |---|---|---|---|---|---|---|---|---|---|
> > | AFM | 0.68 | 0.53 | 0.52 | 0.45 | 0.42 | 0.44 | 0.38 | 0.38 | 0.46 |
> > | MoLPC | 0.66 | 0.59 | **0.60** | **0.51** | 0.47 | 0.42 | 0.35 | 0.36 | 0.50 |
> > | **Ours** | **0.74** | **0.69** | 0.56 | 0.48 | **0.49** | **0.46** | **0.44** | **0.42** | **0.54** |
> >
> > Setting: "ESMFold Dimer"
> > | Chain Number | 3 | 4 | 5 | 6 | 7 | 8 | 9 | 10| overall |
> > |---|---|---|---|---|---|---|---|---|---|
> > | AFM | **0.68** | 0.53 | 0.52 | **0.45** | **0.42** | 0.44 | 0.38 | 0.38 | 0.46 |
> > | MoLPC | 0.52 | 0.50 | 0.41 | 0.39 | 0.36 | 0.27 | 0.33 | 0.33 | 0.39 |
> > | **Ours** | **0.68** | **0.60** | **0.55** | **0.45** | 0.39 | **0.46** | **0.42** | **0.42** | **0.49** |
> >
> > We will add these meaningful results to our revised manuscript.
> >
> > **[Cons. 4 Missing references]** Thank you very much for the valuable references provided by the reviewer. We will add them to the revised manuscript PDF, which is helpful in improving the quality of our manuscript.
> >
> > **[Q1 The Alphafold-multimer baseline]** We respectfully clarify that our method is a **framework to predict the best possible protein multimers (>>2 protein chains) that takes as input dimer structure prediction methods (=2 protein chains)**. Among those dimer structure prediction methods, some ("HDock setting") will take ground-truth monomer structures as input and some ("AFM setting" and "ESMFold setting") will not. **However, under all three of these settings, our method can consistently outperform other baselines.**
> >
> >
> > **[Q2 Use predicted monomers]** Thank you for the valuable suggestions. **To obtain pre-calculated dimer structures, the "AFM Dimer" and "ESMFold Dimer" settings both take amino acid sequences as input.** The experiments show that our method can work well with these dimer prediction methods and achieve considerable accuracy.
> > Although the dimer prediction methods can affect the final accuracy of our method, our method outperforms AlphaFold-Multimer and MoLPC across all settings.
> >
> >
> > **[re1]** Bryant P, Pozzati G, Zhu W, et al. Predicting the structure of large protein complexes using AlphaFold and Monte Carlo tree search. Nature communications, 2022, 13(1): 6028.
> >
> > **[re2]** Luo Y, Li S, Sun Y, et al. xTrimoDock: Rigid Protein Docking via Cross-Modal Representation Learning and Spectral Algorithm. bioRxiv, 2023: 2023.02. 06.527251.
> >
> > **[re3]** Wang R, Li S, Wang X, et al. xTrimoDock: Cross-Modal Transformer for Multi-Chain Protein Docking. Openreview, 2022.
> >
> > **[re4]** Wang R, Sun Y, Luo Y, et al. Injecting Multimodal Information into Rigid Protein Docking via Bi-level Optimization. NeurIPS, 2023.

---

> > > ### Author Response · Authors · 2023-11-23
> > > **We are glad to have a further discussion**
> > >
> > > Dear reviewer PqHe,
> > >
> > > As the discussion period is nearing its end, we would appreciate it if you could confirm whether our responses have addressed your concerns. If you have any additional comments, we are more than willing to engage in further discussion. To aid your understanding, we have clarified essential information regarding our problem setting and involved baselines. Kindly refer to our detailed responses for further information.
> > >
> > > Thank you very much for your time and consideration.

---

### Meta-Review · Area_Chair_cZ5g · 2023-12-08

**Metareview:**

The paper proposes a pre-training and prompt tuning framework for multimer structure prediction.

The AC and reviewers all agree that the approach is interesting and the problem relevant.

While several reviewers found the presentation of the initial submission confusing, the revised manuscript is a nice improvement.

We urge the authors to incorporate into their manuscript the tables they provided in their feedback to reviewers PqHe and 6yks on performance vs different numbers of chains.

**Justification For Why Not Higher Score:**

The proposed solution combines several existing pieces, and the reader unfamiliar with the problem might need to check extensive background materials to appreciate the contributions. This in part explains why several reviewers found the initial submission confusing. Fortunately the authors have done a great job at improving clarity.

**Justification For Why Not Lower Score:**

The paper considers an interesting problem. The proposed approach is sound, exhibits good performance, and results in pertinent insights. The authors have satisfactorily addressed all the reviewers concerns.

---

### Decision · Program_Chairs · 2024-01-16

Accept (poster)